# STREAMLINING REDUNDANT LAYERS TO COMPRESS LARGE LANGUAGE MODELS

**Xiaodong Chen**[2,3†]**, Yuxuan Hu**[2,3†]**, Jing Zhang**[1,2*]**, Yanling Wang**[4]**, Cuiping Li**[1,2]**, Hong Chen**[1,2]
[1] Engineering Research Center of Database and Business Intelligence, MOE, China
[2] School of Information, Renmin University of China,Beijing, China
[3] Key Laboratory of Data Engineering and Knowledge Engineering, MOE, China
[4] Zhongguancun Laboratory, China
`{chenxiaodong,huyuxuan1999,zhang-jing,licuiping,chong}@ruc.edu.cn`
`wangyl@zgclab.edu.cn`

## ABSTRACT

This paper introduces LLM-Streamline, a pioneer work on layer pruning for large language models (LLMs). It is based on the observation that different layers have varying impacts on hidden states, enabling the identification of less important layers to be pruned. LLM-Streamline comprises two parts: layer pruning, which removes consecutive layers with the lowest importance based on target sparsity, and layer replacement, a novel module that trains a lightweight network to replace the pruned layers to mitigate performance loss. Additionally, a new metric called stability is proposed to address the limitations of the widely used accuracy metric in evaluating model compression. Experiments show that LLM-Streamline outperforms both previous and concurrent state-of-the-art pruning methods in terms of both performance and training efficiency. Our code is available at this repository.

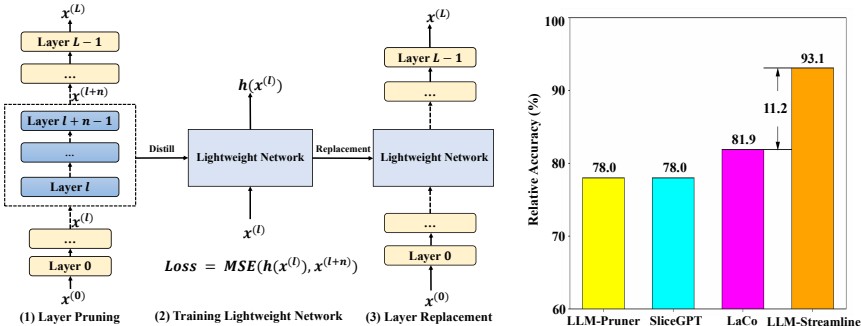

Figure 1: The left side of the figure illustrates the LLM-Streamline workflow, which includes layer pruning to remove consecutive layers and layer replacement where a lightweight network is trained to replace the pruned layers. The right side of the figure presents the comparison results of LLM-Streamline with the state-of-the-art (SOTA) pruning methods on 12 classification benchmarks (details in Section 4.2) after pruning about 25% of the parameters on Llama2-7B. LLM-Streamline achieves 11.2% higher relative accuracy than these methods, where the relative accuracy represents the percentage of the original model's accuracy retained by the pruning method.

## 1 INTRODUCTION

Large language models (LLMs) built on the Transformer architecture (Vaswani et al., 2017) have gained widespread attention and are applied across diverse domains and tasks. However, as LLMs increase in size, their hardware requirements escalate substantially, thereby constraining their applicability and impeding their deployment in real-world scenarios. To reduce the hardware requirements

---

[†]These authors contributed equally.
[*]Jing Zhang is the corresponding author.

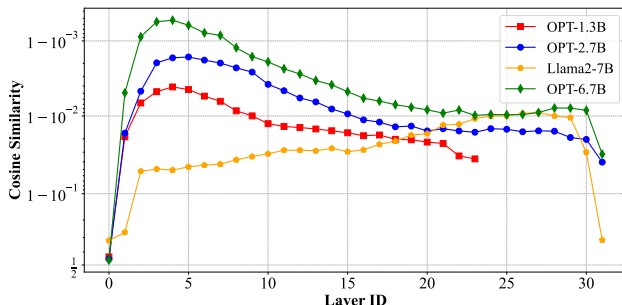

Figure 2: The cosine similarity between the input and output hidden states of each layer in OPT-1.3B, OPT-2.7B, OPT-6.7B, and Llama2-7B.

for deploying LLMs, research efforts have been devoted to developing compact models that maintain high performance through model compression (Zhu et al., 2023; Wang et al., 2024). Currently, model compression techniques are widely categorized into knowledge distillation (Hinton et al., 2015; Gou et al., 2021; Li et al., 2022; Huang et al., 2022; Ho et al., 2022), quantization (Liu et al., 2021; Gholami et al., 2022; Dettmers et al., 2022), and pruning (Louizos et al., 2017; Chen et al., 2023; Frantar & Alistarh, 2023; Das et al., 2023; Sun et al., 2023; Xia et al., 2023). Knowledge distillation achieves compression by transferring the capabilities of a larger teacher model to a smaller student model. Quantization compresses the model by quantizing the weights to lower precision. Alternatively, pruning compresses the model by eliminating unimportant parameters and modules.

This work focuses on the popular pruning methods. Previous approaches for LLM primarily prune dense matrices (Ashkboos et al., 2024), attention heads (Michel et al., 2019; Voita et al., 2019), filters (McCarley et al., 2019; Prasanna et al., 2020), or prune parameters to reduce an LLM's hidden dimension (Xia et al., 2023; van der Ouderaa et al., 2023; Hu et al., 2024). Despite their effectiveness, these methods often result in structural irregularities, making it inflexible to store and deploy the pruned models. In contrast, layer pruning method simply reduces the depth of LLMs. As the layers of LLMs are stored in data structures like nn.ModuleList, layer pruning only requires removing elements from this list, making it more flexible for application. Therefore, exploring an effective layer-wise pruning method is crucial.

The core idea of layer pruning is to identify and remove less important layers in an LLM. Specifically, the effect of each layer can be viewed as a transformation of the hidden states. If the input and output hidden states of a particular layer are highly similar, such as exhibiting high cosine similarity, we can say that the layer has a small impact on adjusting the hidden states. As illustrated in Fig. 2, our pilot study shows that certain contiguous layers indeed have smaller impact on the hidden states, indicating they are less important and suitable for pruning. Some concurrent works (Song et al., 2024; Kim et al., 2024; Yang et al., 2024; Men et al., 2024; Gromov et al., 2024) also explore layer pruning. These studies either prune unimportant layers directly without further training (Song et al., 2024; Men et al., 2024) or fine-tune the pruned model to enhance performance (Kim et al., 2024; Yang et al., 2024; Gromov et al., 2024). Directly removing layers can lead to more performance degradation. While parameter-efficient fine-tuning techniques like LoRA (Hu et al., 2021) are used to train the pruned LLM, fine-tuning the model to make the original non-contiguous layers compensate for the performance degradation is not an easy task (details in Section 2.3).

In this work, we propose a layer pruning method called LLM-Streamline, which exhibits advantages in both performance and training efficiency while requiring less training data. LLM-Streamline comprises two components: layer pruning and layer replacement. According to a certain target sparsity, the first step removes consecutive layers with the lowest importance from the original LLM. Subsequently, we train a lightweight network to replace the pruned layers, aiming to recover the performance degradation caused by pruning. We can employ various architectures for this lightweight network, including a feed-forward neural network (FFN), a SwiGLU-based feed-forward neural network (SwiGLU), and a Transformer layer.

Additionally, we find that existing accuracy metrics for evaluating model compression methods have limitations. Specifically, in natural language understanding tasks that involve multiple-choice

classification, a compressed model may guess correct answers for samples on which the original model was uncertain, resulting in an overestimation of the compression performance. To address this issue, we propose a new metric named stability, which measures the consistency of predictions before and after pruning, considering the prediction confidence of the original model.

Overall, this paper makes the following contributions:

- We propose LLM-Streamline, a layer-wise pruning algorithm that demonstrates superior effectiveness and efficiency compared to concurrent methods. To mitigate the potential performance degradation caused by pruning, we propose to use a lightweight network to approximate the functionality of the pruned layers.

- We propose a new metric called stability, which considers both the prediction confidence of the original model and the consistency of predictions before and after pruning. Stability provides a more accurate reflection of the pruned model's performance in classification tasks compared to the widely used accuracy metric.

- We conduct experiments on 12 well-known classification benchmarks and 3 generation benchmarks. Our results show that for an LLM with 7B or 13B parameters and a 25% pruning rate, we can maintain 93% performance in classification tasks and 77% in generation tasks without requiring a lot of training data, outperforming existing SOTA pruning methods.

## 2 LLM-STREAMLINE

The workflow of the LLM-Streamline framework, shown in Fig. 1 (a), comprises two main steps: layer pruning and layer replacement. First, we prune redundant layers from the LLMs. Then, we train a lightweight network to replace the pruned layers to restore the model's performance.

### 2.1 LAYER REDUNDANCY IN LLMS

LLMs primarily utilize a Transformer architecture, consisting of a series of Transformer decoder layers. These layers adopt a residual structure, so the effect of each Transformer layer can be viewed as a transformation of the input hidden states. Assuming that the parameters of the $\ell$-th layer $f$ are denoted as $\theta^{(\ell)}$, and its input hidden states are represented by $\boldsymbol{x}^{(\ell)}$, the layer $f$ can be expressed as

$$\boldsymbol{x}^{(\ell+1)} = \boldsymbol{x}^{(\ell)} + f(\boldsymbol{x}^{(\ell)}, \theta^{(\ell)}). \tag{1}$$

In Eq. 1, the $\ell$-th layer $f$ contributes a transformation $f(\boldsymbol{x}^{(\ell)}, \theta^{(\ell)})$ to the input $\boldsymbol{x}^{(\ell)}$. Therefore, we assess the importance of each layer in LLMs by evaluating its impact on the input hidden states. We use the cosine similarity $\cos(\cdot, \cdot)$ between input $\boldsymbol{x}^{(\ell)}$ and output $\boldsymbol{x}^{(\ell+1)}$ as a metric. Essentially, a higher cosine similarity between the input and output of a layer indicates lower importance, and vice versa. This interpretation arises from the observation that a high cosine similarity suggests the layer's transformation is minimal, making it more amenable to pruning.

To measure the importance of different layers in LLMs, we randomly select samples from the pre-training data (details in Section 4.1). We then record the hidden states generated by the LLMs for these samples and compute the cosine similarity between the input and output hidden states of each layer. The computation of cosine similarity can be formalized as follows,

$$\cos\left(\boldsymbol{x}^{(\ell)}, \boldsymbol{x}^{(\ell+1)}\right) = \mathbb{E}_{(\boldsymbol{x}_i^{(\ell)}, \boldsymbol{x}_i^{(\ell+1)}) \in \mathcal{D}} \left( \frac{1}{L} \sum_{j=1}^{L} \frac{\boldsymbol{x}_{i,j}^{(\ell)} \cdot \boldsymbol{x}_{i,j}^{(\ell+1)}}{\|\boldsymbol{x}_{i,j}^{(\ell)}\| \cdot \|\boldsymbol{x}_{i,j}^{(\ell+1)}\|} \right), \tag{2}$$

where $\mathcal{D}$ denotes the recorded hidden states from different samples, $\boldsymbol{x}_i^{(\ell)}, \boldsymbol{x}_i^{(\ell+1)} \in R^{d \times L}$ denotes the input and output hidden states of the $i$-th sample respectively, $d$ denotes the hidden size and $L$ denotes the sequence length of each sample.

To mitigate the effects of model size and model structure, we conduct experiments on four models OPT-1.3B, OPT-2.7B, OPT-6.7B (Zhang et al., 2022), and Llama2-7B (Touvron et al., 2023). The results, illustrated in Fig. 2, show that there is high cosine similarity between the input and output of several consecutive layers in all models, indicating a low level of importance.

**Discussion I: Why not use other similarity to measure the importance of layers?** In deep learning, cosine similarity is widely employed to measure the similarity between two vectors (Chen et al., 2020; Chen & He, 2021; Reimers, 2019). Alongside it, dot product and Euclidean distance are also utilized, but they additionally consider vector magnitude. Current research suggests that the hidden states of Transformers with the Pre-Norm architecture tend to grow as the depth of layers increases (Liu et al., 2023). This trend leads to a bias where deeper layers in the model have higher dot product similarity, while earlier layers have smaller Euclidean distances. Consequently, we opt for cosine similarity, which is agnostic to the magnitude of the vectors.

**Discussion II: Why not use perplexity as the metric to measure the importance of layers?** Some concurrent layer pruning work uses perplexity as the metric to measure the importance of layers (Song et al., 2024; Kim et al., 2024). Specifically, they remove each layer one at a time, measuring the change in perplexity of the model on the pre-training data, and eliminate the layer that causes the least change. This process is repeated multiple times to remove several layers. However, we think perplexity is a highly data-sensitive metric, which results in different layers being removed when pruning with different pre-training data. This also results in a situation where, although the perplexity of the pruned model on the pre-training data used for pruning is low, it performs poorly on other datasets. In contrast, the cosine similarity is highly stable and always leads to the same pruned layers on different pre-training data. We conduct detailed experiments in the Appendix A to demonstrate that perplexity is a highly data-sensitive metric and performs poorly on downstream tasks.

## 2.2 LAYER PRUNING

As Fig. 2 shows, the less important layers are often contiguous. Hence, given number of pruned layers $n$ determined by a target sparsity, we remove $n$ contiguous layers by finding the initial layer $\ell^*(n)$ corresponding to the highest cosine similarity for pruning:

$$\ell^*(n) = \arg\max_{\ell} \cos(\boldsymbol{x}^{(\ell)}, \boldsymbol{x}^{(\ell+n)}), \tag{3}$$

where we randomly select samples from the pre-training data to compute the cosine similarity between $\boldsymbol{x}^{(\ell)}$ and $\boldsymbol{x}^{(\ell+n)}$, as outlined in Eq. 2.

## 2.3 LAYER REPLACEMENT

After the layer pruning process, we aim to replace the pruned layer with a lightweight network that has much fewer parameters. The rationale is that these layers contribute only minor transformations to the input. Therefore, we hypothesize that the cumulative effect of these layers can be approximated by a lightweight network. Specifically, after identifying the initial layer $\ell^*(n)$ for pruning, we use $(\boldsymbol{x}^{(\ell^*)}, \boldsymbol{x}^{(\ell^*+n)})$ as the training data to train the lightweight network using mean squared error (MSE) loss, which can be formalized as follows:

$$\min_{h} \mathbb{E}_{(\boldsymbol{x}_i^{(\ell^*)}, \boldsymbol{x}_i^{(\ell^*+n)}) \in \mathcal{D}} \text{MSE}(h(\boldsymbol{x}_i^{(\ell^*)}), \boldsymbol{x}_i^{(\ell^*+n)}), \tag{4}$$

where $h$ denotes the lightweight network, $\mathcal{D}$ denotes the recorded hidden states of samples.

**Discussion: Layer Replacement or Fine-Tuning Pruned LLMs?** Here, we discuss why opt for layer replacement, instead of using common Parameter-Efficient Fine-Tuning (PEFT) methods such as LoRA (Hu et al., 2021) and QLoRA (Dettmers et al., 2023) after layer pruning.

First, from the perspective of resource overhead, layer replacement is more adaptable to hardware resource constraints compared to other methods. Fine-tuning the model using the PEFT methods requires storing the model's weights, activation values, and the optimizer state of the PEFT module in the GPU during training. In contrast, layer replacement involves two stages: dataset construction and model training. The first stage only requires storing the model's weight and the forward computation overhead, and the second stage only requires storing of the lightweight network's weight, activation values of lightweight network, and the optimizer state of lightweight network. Therefore, layer replacement can also be implemented under conditions of hardware resource constraints.

Second, layer replacement uses a lightweight network to replace the pruned layer, and distills the knowledge of the pruned layer into the lightweight network using the MSE loss function. Unlike layer replacement, we speculate that training the model after pruning with LoRA is a process of

Table 1: (a) Number of samples in TP, FN, FP, and TN. (b) The PPL standard deviation results ($\times 10^{-3}$) for Llama2-7B and its pruned version on Race-H.

| Dataset | #TP | #FN | #FP | #TN |
|---|---|---|---|---|
| C3 | 543 | 257 | 210 | 815 |
| CHID | 269 | 563 | 177 | 993 |
| Race-M | 380 | 95 | 129 | 832 |
| Race-H | 938 | 305 | 353 | 1902 |

| Model | TP | FN | FP | TN |
|---|---|---|---|---|
| Llama2-7B | 1.12 | 0.87 | 0.94 | 1.02 |
| w/ pruning | 1.13 | 0.84 | 0.88 | 0.92 |

redistributing the function of the pruned layers across the remaining layers. Therefore, substituting the pruned layers with a lightweight network could be a less challenging training task than redistributing the function of the pruned layers across the remaining layers. In the experiments of Section 4.7, we demonstrate that layer replacement has better performance compared to LoRA.

## 3 METRICS FOR EVALUATING PRUNED MODELS

Accuracy is the most commonly used metric for evaluating LLMs in classification tasks. However, accuracy may overestimate the performance of the model after compression, since it does not take into account the consistency of the model's answers before and after compression. In this section, we analyze such limitation and propose a novel metric for evaluating compressed models.

### 3.1 SHORTCOMING OF ACCURACY METRIC

When evaluating the natural language understanding capabilities of LLMs, most existing benchmarks frame the task as a classification task. A classification task with $k$ choices and comprising $N$ samples is denoted as $\mathcal{T} = \{(x_i, c_{i,1}, c_{i,2}, ..., c_{i,k}, y_i)\}_{i=1}^{N}$, where $x_i$ represents the question in the $i$-th sample, $c_{i,j}$ represents the $j$-th choices, and $y_i$ represents the correct choice. The input to the classification task consists of a question accompanied by multiple choices, and the LLM is required to select the correct answer from these choices. Typically, each choice is concatenated with the question to form multiple sentences, and the perplexity (PPL) of each sentence is computed. The choice corresponding to the sentence with the lowest PPL is selected as the answer.

Typically, model pruning results in decreased model performance. However, when we evaluate the model pruned by the method described in Sec 2.2, we unexpectedly observe the accuracy of the pruned model has been improved on some classification tasks. We define $\mathcal{M}$ to denote the original LLM, $\bar{\mathcal{M}}$ to denote the compressed LLM, and $\hat{y}(\mathcal{M})$ to denote the choice predicted by the model $\mathcal{M}$. To further investigate this phenomenon, we analyze the experimental results using the confusion matrix (Li et al., 2024). Specifically, we count the number of samples and average standard deviation (std) for the PPL of the samples for each term of the confusion matrix. The calculation of the std for the PPL of the $i$-th sample is defined as follows:

$$\text{PPL}_{i,j} = \text{PPL}(\mathcal{M}(x_i, c_{i,j})), \text{PPL}_i = \frac{\sum_{j=1}^{k} \text{PPL}_{i,j}}{k}, \text{std}_i = \sqrt{\frac{\sum_{j=1}^{k}(\text{PPL}_{i,j} - \text{PPL}_i)^2}{k-1}}, \quad (5)$$

where $\text{PPL}_{i,j}$ denotes the PPL for the sentence created by question $x_i$ and choice $c_{i,j}$ of the model before pruning, $\text{std}_i$ denotes the std for PPL of the $i$-th sample. A higher $\text{std}_i$ value indicates the LLM exhibits greater confidence in answering the question $x_i$.

Each term of the confusion matrix is defined as follows,

- TP $\left[\hat{y}_i(\mathcal{M}) = y_i \wedge \hat{y}_i(\bar{\mathcal{M}}) = y_i\right]$ is a set of samples where the model answers correctly both before and after pruning.
- FN $\left[\hat{y}_i(\mathcal{M}) = y_i \wedge \hat{y}_i(\bar{\mathcal{M}}) \neq y_i\right]$ is a set of samples where the model answers correctly before pruning but incorrectly after pruning.
- FP $\left[\hat{y}_i(\mathcal{M}) \neq y_i \wedge \hat{y}_i(\bar{\mathcal{M}}) = y_i\right]$ is a set of samples where the model answers incorrectly before pruning but correctly after pruning.
- TN $\left[\hat{y}_i(\mathcal{M}) \neq y_i \wedge \hat{y}_i(\bar{\mathcal{M}}) \neq y_i\right]$ is a set of samples where the model answers incorrectly both before and after pruning.

Table 1 presents the counts of samples in TP, FN, FP, TN in sevaral datasets, and also the PPL standard deviation in Race-H dataset. We can observe that the std for TP and TN is significantly higher than that for FN and FP. This indicates that the model is more uncertain about the FN and FP samples. In addition, the samples in FP constitute a considerable proportion of the total samples, implying that the model may guess the correct answer for a significant portion after pruning. This phenomenon suggests that the accuracy metric may overestimate the performance of the compressed model.

## 3.2 STABILITY METRIC

We propose a novel metric stability to evaluate the performance of LLMs after pruning, i.e.,

$$\text{Stability}(\mathcal{M}, \bar{\mathcal{M}}) = \frac{\sum_{i=1}^{N} \left( \exp\left(\text{std}_i\right) \cdot \mathbb{1}_{[i \in \text{TP} \cup \text{TN}]} \right)}{\sum_{i=1}^{N} \exp\left(\text{std}_i\right)}, \tag{6}$$

where the identifier $\mathbb{1}_{[i \in \text{TP} \cup \text{TN}]}$ is used to indicate whether the $i$-th sample belongs to TP and TN. We use $\text{std}_i$ as the weight of the $i$-th sample. Because the $\text{std}$ of different samples varies significantly, to mitigate the influence of samples with excessively large standard deviations, we apply the $\exp$ function to moderate the weight differences among samples. Different from accuracy, stability focuses on the model's confidence in its answers and the consistency between the model before and after pruning on a task, aligning more closely with the goal of model pruning, i.e., ensuring the pruned model remains as similar as possible to the original model.

## 4 EXPERIMENTS

In this section, we first compare our proposed method, LLM-Streamline, with several popular pruning methods to demonstrate its effectiveness (4.4). Next, we analyze the impact of different sizes and structures of lightweight networks on model performance (4.5) and evaluate performance under various pruning ratios (4.6). Finally, we compare our layer replacement approach with the well-known PEFT method, LoRA (Hu et al., 2021) (4.7), showing that layer replacement offers superior performance and reduced memory overhead.

## 4.1 SETUP

We conduct experiments on popular open-source LLMs, including Llama2-7B and Llama2-13B (Touvron et al., 2023). Following previous work (Men et al., 2024; Yang et al., 2024), we perform experiments with a 25% pruning rate and extract data from pre-training dataset SlimPajama which contains data from different domains for layer pruning and layer replacement. Sheared LLaMa (Xia et al., 2023) finds that the performance degradation of pruned models varies across different domains, and proposes determining the distribution of data from different domains based on the degree of performance degradation. Therefore, we randomly sample the data based on the distribution used by Sheared LLaMa (Xia et al., 2023), finally constructing the dataset containing 30,000 pieces of data. We randomly select 500 samples from this dataset and input them into LLMs, generating Fig. 2, and use these 500 data samples for layer pruning. All 30,000 pieces of data are used to train the lightweight network. We utilize two types of lightweight networks: a Feed-Forward Neural Network (FFN), referred to as Ours (FFN), and a Transformer Layer, referred to as Ours (Layer). The FFN is randomly initialized, while the Transformer Layer inherits the parameters from the first pruned layer. Additionally, we explore a purely pruning approach without incorporating any lightweight network, denoted as Ours (None). Further experimental details are available in the Appendix C.1.

## 4.2 BENCHMARK

We use 12 natural language understanding benchmarks for evaluation: **CMNLI** (Xu et al., 2020), **HellaSwag**(HeSw) (Zellers et al., 2019), **PIQA** (Bisk et al., 2020),**CHID** (Zheng et al., 2019), **WSC** (Levesque et al., 2012),**CommonSenseQA**(CoQA) (Talmor et al., 2018), **BoolQ** (Clark et al., 2019),**MMLU** (Hendrycks et al., 2020), **CMMLU** (Li et al., 2023),**Race-High/Middle** (Lai et al., 2017), **C3** (Sun et al., 2020). The tasks in these benchmarks are formalized as classification tasks, so we refer to these benchmarks as classification benchmarks. For these benchmarks, we use both

Table 2: Accuracy of pruning methods on classification benchmarks. "*" indicates that we refer to the results in the original paper. Retained performance (RP) represents the percentage of the original model's performance retained by the pruning method.

| LLM | Method | Ratio | Benchmarks | | | | | | | | | | | | Average | RP |
|---|---|---|---|---|---|---|---|---|---|---|---|---|---|---|---|---|
| | | | C3 | CMNLI | CHID | BoolQ | WSC | CoQA | HeSW | PIQA | Race-M | Race-H | MMLU | CMMLU | | |
| Llama2-7B | Dense | 0.00% | 43.8 | 33.0 | 41.6 | 70.8 | 37.5 | 66.7 | 71.3 | 78.1 | 33.1 | 35.5 | 46.8 | 31.8 | 49.2 | 100.0 |
| | LLMPruner | 24.8% | 29.7 | 33.4 | 28.4 | 58.7 | 40.4 | 48.5 | 54.6 | 72.0 | 22.9 | 22.0 | 25.3 | 25.0 | 38.4 | 78.0 |
| | SliceGPT | 25.4% | 31.5 | 31.6 | 18.5 | 59.9 | 43.3 | 49.6 | 47.5 | 68.3 | 27.0 | 29.4 | 28.8 | 24.8 | 38.4 | 78.0 |
| | LaCo* | 27.0% | 39.7 | 34.4 | 36.1 | 64.1 | 40.4 | 45.7 | 55.7 | 69.8 | 23.6 | 22.6 | 26.5 | 25.2 | 40.3 | 81.9 |
| | Ours (None) | 24.0% | 40.2 | 34.4 | 21.5 | 67.3 | 40.4 | 51.7 | 59.7 | 69.0 | 35.2 | 34.7 | 44.6 | 28.9 | 44.0 | 89.4 |
| | Ours (FFN) | 25.0% | 40.7 | 33.0 | 22.8 | 65.9 | 38.5 | 60.6 | 61.2 | 71.2 | 38.0 | 38.7 | 47.0 | 31.7 | 45.8 | 93.1 |
| | Ours (Layer) | 24.0% | 43.3 | 33.0 | 24.1 | 67.5 | 36.5 | 59.2 | 61.1 | 71.5 | 34.8 | 37.0 | 45.5 | 29.4 | 45.2 | 91.9 |
| Llama2-13B | Dense | 0.00% | 47.5 | 33.0 | 47.2 | 71.5 | 51.0 | 66.8 | 74.8 | 79.8 | 60.0 | 58.1 | 55.8 | 38.7 | 57.0 | 100.0 |
| | LLMPruner | 24.4% | 29.5 | 33.0 | 29.5 | 58.0 | 47.1 | 43.7 | 54.7 | 72.7 | 21.9 | 22.5 | 25.2 | 24.9 | 38.6 | 67.7 |
| | SliceGPT | 23.6% | 38.6 | 30.5 | 18.3 | 37.8 | 42.3 | 38.3 | 45.6 | 61.9 | 24.0 | 25.0 | 30.6 | 25.6 | 34.9 | 61.2 |
| | LaCo* | 24.6% | 44.9 | 32.9 | 40.1 | 64.0 | 52.9 | 52.7 | 64.4 | 74.3 | 56.6 | 54.5 | 45.9 | 32.6 | 51.3 | 90.0 |
| | Ours (None) | 24.6% | 47.0 | 33.0 | 36.5 | 62.3 | 64.4 | 58.8 | 66.6 | 73.5 | 60.2 | 58.3 | 54.8 | 38.4 | 54.5 | 95.6 |
| | Ours (FFN) | 25.4% | 45.8 | 33.0 | 37.1 | 67.4 | 37.5 | 64.4 | 67.9 | 74.0 | 58.6 | 58.2 | 55.7 | 38.6 | 53.2 | 93.3 |
| | Ours (Layer) | 24.6% | 45.7 | 33.0 | 38.0 | 66.2 | 36.5 | 63.8 | 69.1 | 75.1 | 58.0 | 57.4 | 55.1 | 39.2 | 53.1 | 93.2 |

Table 3: Stability of pruning methods on classification benchmarks. The stability of the original model is 1.0, because stability is measured by comparing the prediction results of the original model.

| LLM | Method | Ratio | Benchmarks | | | | | | | | | | | | Average |
|---|---|---|---|---|---|---|---|---|---|---|---|---|---|---|---|
| | | | C3 | CMNLI | CHID | BoolQ | WSC | CoQA | HeSW | PIQA | Race-M | Race-H | MMLU | CMMLU | |
| Llama2-7B | LLMPruner | 24.8% | 72.8 | 94.0 | 74.1 | 70.8 | 87.5 | 71.0 | 79.9 | 86.8 | 52.4 | 55.2 | 53.3 | 65.9 | 72.0 |
| | SliceGPT | 25.4% | 53.2 | 35.4 | 53.3 | 77.1 | 80.8 | 75.3 | 71.6 | 78.7 | 90.7 | 85.3 | 60.3 | 56.7 | 68.2 |
| | Ours (None) | 24.0% | 76.6 | 38.7 | 65.3 | 81.4 | 87.5 | 74.7 | 80.7 | 81.0 | 73.7 | 67.9 | 80.1 | 70.8 | 73.2 |
| | Ours (FFN) | 25.0% | 79.8 | 100 | 64.1 | 83.1 | 93.3 | 80.7 | 84.7 | 84.6 | 85.1 | 79.0 | 87.5 | 82.5 | 83.7 |
| | Ours (Layer) | 24.0% | 79.8 | 100 | 64.4 | 86.3 | 95.2 | 81.7 | 85.3 | 85.6 | 81.8 | 79.0 | 82.4 | 71.0 | 82.7 |
| Llama2-13B | LLMPruner | 24.4% | 71.6 | 100 | 69.2 | 70.5 | 65.4 | 69.5 | 77.8 | 86.7 | 42.3 | 35.6 | 48.1 | 52.3 | 65.8 |
| | SliceGPT | 23.6% | 62.2 | 39.5 | 51.4 | 27.1 | 68.3 | 65.5 | 64.9 | 75.6 | 45.3 | 43.4 | 52.7 | 53.9 | 54.1 |
| | Ours (None) | 24.6% | 84.2 | 99.9 | 71.8 | 77.4 | 46.2 | 82.2 | 85.7 | 86.5 | 83.3 | 83.6 | 89.1 | 83.8 | 81.1 |
| | Ours (FFN) | 25.4% | 85.7 | 100 | 72.5 | 79.8 | 59.6 | 89.2 | 89.4 | 89.7 | 84.8 | 83.3 | 93.6 | 90.7 | 84.9 |
| | Ours (Layer) | 24.6% | 87.4 | 100 | 74.1 | 81.3 | 58.6 | 89.0 | 90.5 | 90.5 | 84.2 | 83.0 | 92.5 | 85.5 | 84.7 |

accuracy and stability as metrics for evaluating the models. Additionally, we include 3 benchmarks: **XSum** (Narayan et al., 2018), **GSM8K** (Cobbe et al., 2021) and **StrategyQA** (Geva et al., 2021), to demonstrate the LLM's performance on generation tasks after pruning. We refer to these tasks as generation benchmarks. Following the evaluation framework of OpenCompass (Contributors, 2023), we use accuracy to evaluate **StrategyQA** and **GSM8K**, and use ROUGE1 to evaluate **Xsum**.

## 4.3 BASELINE

We compare several pruning methods that prune the attention heads, the filters of the FFN layer, and the hidden dimension, as well as the concurrent layer-pruning methods LaCo. In addition, **ShortGPT** and **UIDL** (Men et al., 2024; Gromov et al., 2024) can be considered as a variant of our approach, i.e., Ours (None). We also discuss layer pruning methods which use perplexity as the metric, such as **SLEB** (Song et al., 2024), in Appendix A.

**LLM-Pruner** (Ma et al., 2023) prunes attention heads, FFN layer filters, and hidden dimensions by using gradients and activations to estimate the importance of these modules.

**SliceGPT** (Ashkboos et al., 2024) prunes hidden dimensions. It inserts dimensionality reduction matrices into the model and employs Principal Component Analysis (PCA) to initialize and compress the matrices, and then merge them with the original weight matrix to reduce the model's size.

**LaCo** (Yang et al., 2024) prunes layers by dividing the layers into groups, each consisting of multiple consecutive layers, and compresses them separately, whereas our method simply compresses a piece of consecutive layers. LaCo merges consecutive layers by averaging their parameters whereas we train an additional lightweight network to replace these layers.

## 4.4 MAIN RESULTS

We present accuracy and stability for different methods on the classification benchmarks in Table 2 and Table 3, respectively, and Table 4 for the generative benchmarks. The results demonstrate that our proposed LLM-Streamline consistently outperforms the baseline methods. Specifically, in classification tasks, LLM-Streamline surpasses LLM-Pruner by 7% in accuracy and 12% in stability on Llama2-7B, and by 16% in accuracy and 19% in stability on Llama2-13B. LLM-Streamline also surpasses LaCo by 5% in accuracy on Llama2-7B. For generation tasks, LLM-Streamline retains

Table 4: Evaluations on generation benchmarks. "*" indicates that we refer to the results in the original paper.

| LLM | Method | Ratio | Benchmarks Xsum | GSM8K | StrategyQA | Average | RP |
|-----|--------|-------|------|-------|------------|---------|-----|
| Llama2-7B | Dense | 0.00% | 19.4 | 16.5 | 60.2 | 32.0 | 100.0 |
| | LLMPruner | 24.8% | 16.4 | 0.61 | 44.2 | 20.4 | 63.8 |
| | SliceGPT | 25.4% | 12.4 | **3.34** | 45.7 | 20.5 | 64.1 |
| | LaCo* | 27.1% | 15.6 | - | - | - | - |
| | Ours (None) | 24.0% | 14.8 | 1.97 | 41.8 | 19.5 | 60.9 |
| | Ours (FFN) | 25.0% | 18.6 | 2.16 | 46.5 | 22.4 | 70.0 |
| | Ours (Layer) | 24.0% | **20.2** | 1.82 | **52.1** | **24.7** | **77.2** |
| Llama2-13B | Dense | 0.00% | 23.7 | 29.0 | 58.1 | 36.9 | 100.0 |
| | LLMPruner | 24.4% | 17.5 | 1.9 | 43.7 | 21.0 | 56.9 |
| | SliceGPT | 23.6% | 5.0 | 1.9 | 38.3 | 15.1 | 40.9 |
| | LaCo* | 24.6% | 14.5 | - | - | - | - |
| | Ours (None) | 24.6% | 17.7 | 2.35 | 46.0 | 22.0 | 59.6 |
| | Ours (FFN) | 25.4% | 21.4 | 4.10 | **59.6** | **28.4** | **77.0** |
| | Ours (Layer) | 24.6% | **21.8** | **4.70** | 57.3 | 27.9 | 75.6 |

Table 5: Comparison of different lightweight networks on classification benchmarks in terms of average accuracy and stability metrics, where "†" indicates that the intermediate size of the added lightweight network is half that of the default LLM's intermediate size.

| | Layer-Random | Layer-First | Layer-Last | Layer-Avg | FFN$^\dagger$ | FFN | SwiGLU$^\dagger$ | SwiGLU |
|---|---|---|---|---|---|---|---|---|
| Accuracy | 45.1 | 45.2 | 45.6 | 44.4 | **46.0** | 45.8 | 43.8 | 44.2 |
| Stability | 81.2 | 82.7 | 81.9 | 79.2 | 80.7 | **83.7** | 82.6 | 83.3 |

nearly 77% of Llama2-7B and Llama2-13B's capabilities, significantly outperforming other pruning methods. We find that almost all of the pruning methods fail on the GSM8K dataset. However, sufficient training can gradually restore the model's performance on math tasks, and the specific experimental results are shown in Table 28 of Appendix E.8.

Additionally, comparing the average stability (Average) in Table 3 with the retrained performance (RP) in Table 2 reveals that stability is often much lower than accuracy. We also observe that accuracy on Race-M and Race-H even increases after model pruning. Furthermore, we find that without using any lightweight network, Llama2-13B achieves the highest accuracy on classification benchmarks, but its stability on classification benchmarks and performance on generation benchmarks are lower. These results indicate that pruned models tend to make correct guesses on some classification questions that they are uncertain, highlighting the limitations of accuracy as a sole measure of pruning method performance. We also conduct experiments on OPT-1.3B, OPT-2.7B, OPT-6.7B, Baichuan-7B, Baichuan-13B, Baichuan2-7B, Baichuan2-13B (Yang et al., 2023), Llama3.1-8B, Llama3.1-70B (Dubey et al., 2024) and Mixtral-8x7B-v0.1 (Jiang et al., 2024). Details can be found in Appendix E.1, Appendix E.2, Appendix E.3, Appendix E.4, Appendix E.5 and Appendix E.6. In addition, we compare the performance of LLM-Streamline with other methods at a higher pruning ratio of approximately 50%, and the results can be found in Appendix E.7.

## 4.5 IMPACT OF DIFFERENT LIGHTWEIGHT NETWORKS

**While FFN achieves the best result, Transformer layer still has performance potential.** We perform experiments with Llama2-7B using various lightweight network, including Feed-Forward Neural Networks (FFN), SwiGLU-based Feed-Forward Neural Networks (SwiGLU), and Transformer layers. We also explore various initialization methods for the Transformer Layer, including random initialization (Layer-Random), inheritance of the first pruned layer (Layer-First), inheritance of the last pruned layer (Layer-Last), and averaging the pruned layers (Layer-Avg). The average accuracy and stability metrics across all the classification benchmarks are presented in Table 5, with detailed results on each benchmark available in Appendix E.9. The results show that FFN achieves the best results. Meanwhile, for the Transformer Layer, inheriting the pruned first layer yields the best results. In contrast, the performance of Layer-Avg, inspired by LaCo, shows that averaging weights does not achieve the same effectiveness as the pruned first layer. In addition, we plot the validation loss curves for different lightweight networks, as shown in Fig. 3. We can observe that FFN and SwiGLU have already converged by the 10th epoch, whereas the loss of Transformer Layer is still decreasing. This

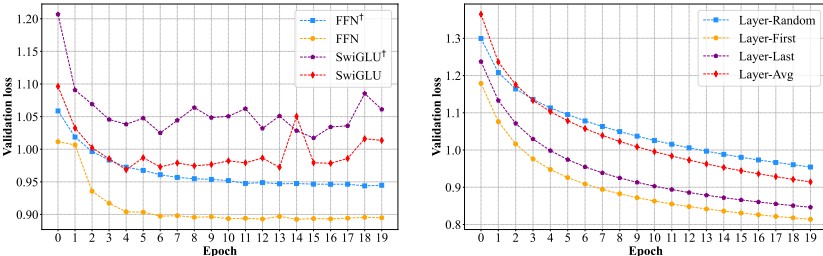

Figure 3: Validation loss curves during training of (a) FFN and SwiGLU; (b) Transformer layer.

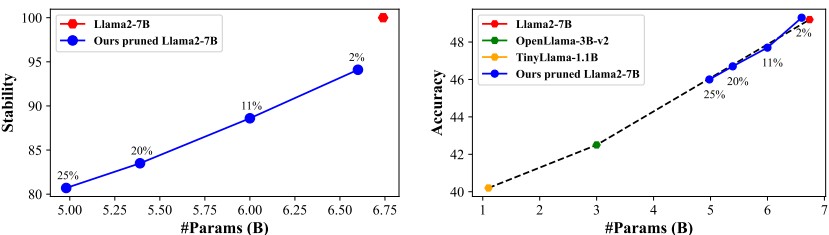

Figure 4: (a) Stability of the pruned Llama2-7B at different pruning ratios. (b) Accuracy of the pruned Llama2-7B at different pruning ratios, compared to the original Llama2-7B, OpenLlama-3B-v2, and TinyLlama-1.1B. Metrics are averaged across classification benchmarks.

indicates that the Transformer layer still has potential, and further training could yield better results, but this would require more computing resources.

## 4.6 IMPACT OF DIFFERENT PRUNING RATIOS

**The performance of the pruned model is linearly correlated with the number of parameters at modest pruning ratios.** To verify the model's performance at various modest pruning ratios, we evaluate our method not only at the approximately 25% pruning ratio but also at ratios of around 2%, 11%, and 20% on Llama2-7B. The average stability and accuracy metrics across all the classification benchmarks are shown in Fig. 4, with details on each benchmark presented in Appendix E.10. By comparing the performance of the original Llama2-7B, TinyLlama-1.1B, OpenLlama-3B-v2, and Llama2-7B pruned at various ratios, we observe a linear correlation between the performance of both the pruned models and the pre-trained original models relative to the number of parameters. This suggests that the performance of models pruned using our method is comparable to that of pre-trained models with the same number of parameters.

## 4.7 COMPARISON OF LAYER REPLACEMENT AND LORA

**Layer Replacement outperforms LoRA in both performance and GPU memory consumption.** We compare the performance of layer replacement with LoRA. Since layer replacement is trained based on hidden states with a different training objective than LoRA, we additionally train one epoch using the language model loss for layer replacement when comparing it with LoRA. The training details can be found in the Appendix C.2. For layer replacement, we freeze the original model's weights and train only the lightweight network. In the case of LoRA, we set the rank to 128 to align the number of parameters trained with those of the lightweight networks. We randomly extract 30,000 samples from SlimPajama-6B for layer replacement training and also test with the entire dataset to evaluate the limited impact of extensive data on performance (details in Appendix E.8). For LoRA, we use 300,000 samples from SlimPajama-6B. Table 6 presents the average accuracy and stability across all classification benchmarks, with detailed results available in Appendix E.11. The findings indicate that layer replacement surpasses LoRA in both accuracy and stability, while also requiring significantly less GPU memory and training data.

Table 6: Comparison of layer replacement and LoRA on classification benchmarks in terms of average accuracy metrics across all benchmarks, where "†" indicates that the intermediate size of the added lightweight network is half that of the default LLM's intermediate size.

| | Layer-First | Layer-Last | Layer-Avg | FFN† | FFN | SwiGLU† | SwiGLU | LoRA |
|---|---|---|---|---|---|---|---|---|
| Accuracy | 46.7 | **46.8** | 46.2 | 45.8 | 46.3 | 44.4 | 45.5 | 44.5 |
| Stability | **85.7** | 85.6 | 83.9 | 83.4 | 85.2 | 84.7 | 84.7 | 82.1 |
| GPU Memory (G) | 27.8 | 27.8 | 27.8 | 25.6 | 27.0 | **25.3** | 26.4 | 56.4 |

Table 7: Comparison of concurrent layer pruning methods, with the metric indicating the importance of layers. Shortened Llama consists of two training stages: initial continual pre-training on the SlimPajama dataset, followed by LoRA fine-tuning on the Alpaca dataset.

| Method | Metric | Need Training | Training Data | Data Size | Training Module | Trainig Method |
|---|---|---|---|---|---|---|
| SLEB | Perplexity | No | None | None | None | None |
| ShortGPT | Cosine Similarity | No | None | None | None | None |
| UIDL | Cosine Similarity | Yes | C4 | 164M | LoRA-Adapter | QLoRA |
| LaCO | Cosine Similarity | Yes | Unpublished | 1B | Full Parameters | Fine-tuning |
| Shortened Llama | Taylor Perplexity | Yes | SlimPajama Alpaca | 627B 50k | Full Parameters LoRA-Adapter | Fine-tuning LoRA |
| LLM-Streamline | Cosine Similarity | Yes | SlimPajama | 30k | Lightweight Network | Training Lightweight Network |

## 5 RELATED WORK

Previous pruning methods for LLMs primarily focus on pruning dense matrices (Ashkboos et al., 2024), attention heads (Michel et al., 2019; Voita et al., 2019), filters (McCarley et al., 2019; Prasanna et al., 2020), or hidden dimension (Xia et al., 2023; van der Ouderaa et al., 2023). These approaches often lead to structural irregularities, making pruned models less flexible for deployment. In contrast, layer pruning, which only alters the model's depth, is easier to deploy. Concurrent works in layer pruning alongside LLM-Streamline include LaCo (Yang et al., 2024), ShortGPT (Men et al., 2024), UIDL (Gromov et al., 2024), SLEB (Song et al., 2024), and Shortened Llama (Kim et al., 2024).

LaCo (Yang et al., 2024) divides layers into groups of consecutive layers and compresses them by replacing the consecutive layers with averaged parameter weights. ShortGPT (Men et al., 2024) uses a BI score, equivalent to cosine similarity, to assess layer importance and remove less important layers. Similarly, UIDL (Gromov et al., 2024) uses angular distance, also equivalent to cosine similarity, to determine and remove less important layers, and employs QLoRA to enhance performance. SLEB (Song et al., 2024) calculates layer importance using perplexity and discards those deemed insignificant. Shortened Llama (Kim et al., 2024) explores various layer selection metrics and examines the effectiveness of using continual pre-training and LoRA after pruning. The differences between these layer pruning methods and LLM-Streamline are summarized in Table 7.

Unlike traditional layer pruning methods, LLM-Streamline fundamentally differs by retraining a lightweight model to replace the pruned layers, rather than removing them directly with or without training the pruned model. LLM-Streamline reduces both computation time and resource consumption compared to layer pruning methods (Shortened Llama, LaCo, UIDL) that necessitate retraining. Additionally, LLM-Streamline better preserves the performance of the original LLM compared to concurrent layer pruning methods.

## 6 CONCLUSION

In this paper, we propose LLM-Streamline, a layer pruning-and-replacement algorithm for LLMs. We also identify shortcomings in the existing accuracy metric and propose a new metric called stability for evaluating model compression. Extensive experiments show that this layer replacement method using a lightweight network outperforms previous state-of-the-art pruning methods and demonstrates superior effectiveness and efficiency compared to concurrent layer pruning methods.

## ACKNOWLEDGMENTS

This work is supported by the National Key Research & Develop Plan (2023YFF0725100) and the National Natural Science Foundation of China (62322214, U23A20299, U24B20144, 62172424, 62276270).

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

## A    COMPARISON OF COSINE SIMILARITY AND PERPLEXITY

To demonstrate the sensitivity of perplexity, referencing the SLEB (Song et al., 2024), we prune the Llama2-7B model with different pre-training datasets, including SlimPajama, C4 and wikitext. The experimental results are presented in Table 8. When pruning with cosine similarity, the layers pruned are consistent across different datasets, whereas when pruning with perplexity, the layers vary, indicating the sensitivity of perplexity. In addition, we evaluate the model after pruning with the SlimPajama dataset, and the experimental results are shown in Table 9. This indicates that the model pruned with perplexity shows lower perplexity on the dataset used for pruning, but performs worse on downstream tasks.

Table 8: Pruned layers using perplexity and cosine similarity for pruning.

| LLM | Dataset | Pruned Layers | |
| --- | --- | --- | --- |
| | | **Perplexity** | **Cosine Similarity** |
| Llama2-7B | SlimPajama | 9,10,11,12,21,23,25,27 | 22,23,24,25,26,27,28,29 |
| | wikitext | 9,10,11,12,21,23,24,27 | 22,23,24,25,26,27,28,29 |
| | C4 | 8,9,11,12,22,23,24,25 | 22,23,24,25,26,27,28,29 |

Table 9: Detailed results of accuracy of using perplexity and cosine similarity for pruning. "Perplexity*" refers to the Perplexity of the pruned model on SlimPajama. Using perplexity as the metric can be considered as SLEB, while using cosine similarity as the metric can be considered as a variant of our approach, i.e., Ours (None)(details in Section 4.1).

| LLM | Metric | Perplexity* | C3 | CMNLI | CHID | BoolQ | WSC | CoQA | HeSW | PIQA | Race-M | Race-H | MMLU | CMMLU | Xsum | GSM8k | StrategyQA | Average | RP |
| --- | --- | --- | --- | --- | --- | --- | --- | --- | --- | --- | --- | --- | --- | --- | --- | --- | --- | --- | --- |
| Llama2-7B | Dense | 6.23 | 43.8 | 33.0 | 41.6 | 70.8 | 37.5 | 66.7 | 71.3 | 78.1 | 33.1 | 35.5 | 46.8 | 31.8 | 19.4 | 16.5 | 60.2 | 45.7 | 100.0 |
| | Cosine Similarity | 19.7 | **40.2** | **34.4** | 21.5 | **67.3** | **40.4** | **51.7** | **59.7** | 69.0 | **35.5** | **34.7** | **44.6** | **28.9** | 14.8 | **1.97** | **41.8** | **39.1** | **85.6** |
| | Perplexity | **12.1** | 37.6 | 33.0 | **34.2** | 61.7 | 36.5 | 47.3 | 56.5 | **71.4** | 22.1 | 21.6 | 25.9 | 24.8 | **17.1** | 1.74 | 33.2 | 35.0 | 76.6 |

## B    DATA DISTRIBUTION

We extract the training data from different domains based on the data distribution strategy proposed in Sheared-LLaMa (Xia et al., 2023). The detailed data distribution is shown in Table 10.

Table 10: The proportion of different domains randomly selected from the SlimPajama-6B dataset.

| | **CC** | **GitHub** | **Book** | **StackExchange** | **Wiki** | **ArXiv** | **C4** |
| --- | --- | --- | --- | --- | --- | --- | --- |
| SlimPajama-6B | 54.1% | 4.2% | 3.7% | 2.8% | 3.1% | 3.4% | 28.7% |
| Ours | 36.1% | 0.8% | 9.1% | 1.0% | 3.1% | 0.7% | 49.2% |

## C    TRAINING IMPLEMENTATION DETAILS

### C.1    LIGHTWEIGHT NETWORK TRAINING DETAILS

For both the FFN structure and the SwiGLU structure, the learning rate is set to 1e-3 and the weight decay is 1e-4. For the Transformer layer, the learning rate is set to 1e-5 and the weight decay is 1e-3. The model is trained using a batch size of 32 over 20 epochs. On a single A800 GPU, the training duration for the lightweight network is approximate 5 hours (for the Transformer layer).

### C.2    POST TRAINING DETAILS

For layer replacement, in order to have a fairer comparison with LoRA, we conduct one epoch of post-training with a learning rate of 5e-5, a weight decay of 1e-3, and a batch size of 32. This process takes less than an hour on a single A800 GPU. For LoRA, the model is trained one epochs with a

learning rate of 1e-4, a weight decay of 1e-3, and a batch size of 32. Since the amount of training data used is ten times that of layer replacement, it take approximately 10 hours to complete the training on a single A800 GPU.

## D    INFERENCE SPEED COMPARISON

As shown in the Table 11, we evaluate the inference speed of various pruning methods at similar pruning ratio when generating sequences of length 128. The results indicate that the acceleration effect of LLM-Streamline is slightly inferior to that of SliceGPT and LLM-Pruner.

Table 11: The inference speed of models pruned using different methods.

| Llama2-7B | Dense | LLM-Pruner | SliceGPT | Ours(None) | Ours(FFN) | Ours(Layer) |
|---|---|---|---|---|---|---|
| Pruning Ratio (%) | 0.00 | 24.8 | 25.4 | 24.0 | 25.0 | 24.0 |
| Inference Speed (tokens/s) | 19.87 | 25.91 | 27.20 | 25.68 | 25.88 | 25.68 |

## E    DETAILED EXPERIMENTAL RESULTS

### E.1    EXPERIMENTAL RESULTS OF OPT-6.7B

We conduct experiments on OPT-6.7B. The experimental results are shown in Table 12, Table 13 and Table 14. The results indicate that our proposed LLM-Streamline is superior to the previous SOTA method.

Table 12: Accuracy of different pruning methods on classification benchmarks by pruning OPT-6.7B.

| LLM | Method | Ratio | Benchmarks | | | | | | | | | | | | Average | RP |
|---|---|---|---|---|---|---|---|---|---|---|---|---|---|---|---|---|
| | | | C3 | CMNLI | CHID | BoolQ | WSC | CoQA | HeSW | PIQA | Race-M | Race-H | MMLU | CMMLU | | |
| | Dense | 0.00% | 38.7 | 32.9 | 21.6 | 64.6 | 41.4 | 54.8 | 63.3 | 75.4 | 25.1 | 25.4 | 24.7 | 25.5 | 41.1 | 100 |
| | SliceGPT | 25.6% | **40.0** | 31.2 | **19.5** | 37.9 | 36.5 | 38.2 | 45.6 | 65.8 | **25.8** | **26.0** | **25.8** | 24.8 | 34.8 | 84.7 |
| OPT-6.7B | Ours (None) | 24.0% | 27.6 | **32.5** | 12.7 | 44.8 | 36.5 | 20.6 | 26.5 | 52.1 | 22.1 | 22.4 | 23.6 | **25.2** | 28.9 | 70.3 |
| | Ours (FFN) | 25.0% | 37.6 | 32.1 | 18.7 | **63.7** | 37.5 | 41.8 | **55.9** | 73.2 | 22.7 | 22.2 | 24.4 | 24.9 | 37.9 | 92.2 |
| | Ours (Layer) | 24.0% | 36.4 | 32.0 | 18.9 | 62.4 | **38.5** | **45.1** | 54.3 | **74.0** | 23.6 | 24.2 | 24.3 | **25.2** | **38.2** | **92.9** |

Table 13: Stability of different pruning methods on classification benchmarks by pruning OPT-6.7B.

| LLM | Method | Ratio | Benchmarks | | | | | | | | | | | | Average |
|---|---|---|---|---|---|---|---|---|---|---|---|---|---|---|---|
| | | | C3 | CMNLI | CHID | BoolQ | WSC | CoQA | HeSW | PIQA | Race-M | Race-H | MMLU | CMMLU | |
| | SliceGPT | 25.6% | 66.4 | 39.8 | 73.1 | 30.1 | 79.4 | 75.2 | 73.7 | 82.5 | 72.5 | 69.1 | 68.9 | 67.2 | 66.5 |
| OPT-6.7B | Ours (None) | 24.0% | 56.2 | **62.5** | 71.4 | 41.5 | 87.5 | 48.6 | 50.5 | 57.1 | 63.4 | 62.2 | 62.3 | 62.7 | 60.5 |
| | Ours (FFN) | 25.0% | **74.1** | 36.1 | **77.4** | **74.2** | 90.4 | 82.4 | **88.4** | **92.0** | 68.3 | 63.9 | 74.7 | 65.0 | 73.9 |
| | Ours (Layer) | 24.0% | 72.1 | 35.5 | 76.1 | 72.1 | **91.4** | **83.6** | 87.0 | 91.6 | **72.9** | **71.2** | **78.1** | **70.1** | **75.1** |

Table 14: Accuracy of different pruning methods on generation benchmarks by pruning OPT-6.7B.

| LLM | Method | Ratio | Benchmarks | | | Average | RP |
|---|---|---|---|---|---|---|---|
| | | | Xsum | GSM8K | StrategyQA | | |
| | Dense | 0.00% | 13.4 | 2.2 | 54.3 | 23.3 | 100.0 |
| | SliceGPT | 25.6% | 14.9 | **2.5** | 40.8 | 19.4 | 83.3 |
| OPT-6.7B | Ours (None) | 24.0% | 4.9 | 0 | 0 | 1.6 | 6.87 |
| | Ours (FFN) | 25.0% | 14.8 | 0.8 | 43.6 | 19.7 | 84.5 |
| | Ours (Layer) | 24.0% | **18.4** | **2.5** | **44.4** | **21.8** | **93.6** |

### E.2    EXPERIMENTAL RESULTS OF BAICHUAN-7B AND BAICHUAN-13B

We conduct experiments on Baichuan-7B and Baichuan-13B. The experimental results are shown in Table 15, Table 16 and Table 17. The results indicate that our proposed LLM-Streamline is superior to the previous SOTA method.

Table 15: Accuracy of different pruning methods on classification benchmarks by pruning Baichuan-7B and Baichuan-13B.

| LLM | Method | Ratio | Benchmarks | | | | | | | | | | | | Average | RP |
| --- | --- | --- | C3 | CMNLI | CHID | BoolQ | WSC | CoQA | HeSW | PIQA | Race-M | Race-H | MMLU | CMMLU | | |
| Baichuan-7B | Dense | 0.00% | 55.8 | 35.3 | 91.3 | 61.4 | 39.4 | 58.4 | 65.3 | 77.6 | 29.5 | 30.4 | 43.7 | 43.8 | 52.7 | 100 |
| | LLMPruner | 24.2% | 43.7 | 33.9 | 65.9 | 40.5 | 36.5 | 48.2 | 52.2 | 68.1 | 22.6 | 22.0 | 24.2 | 25.3 | 40.2 | 76.3 |
| | Ours (None) | 24.2% | 33.2 | 32.7 | 25.8 | 60.8 | 36.5 | 36.0 | 34.6 | 58.7 | 22.1 | 21.5 | 25.7 | 38.8 | 35.5 | 67.4 |
| | Ours (FFN) | 25.1% | 53.1 | 36.3 | 69.4 | 53.1 | 36.5 | 48.7 | 53.2 | 69.4 | 23.2 | 24.5 | 37.7 | 39.1 | 45.4 | 86.1 |
| | Ours (Layer) | 24.2% | 55.0 | 36.0 | 77.4 | 48.1 | 36.5 | 49.8 | 54.3 | 69.0 | 22.9 | 23.8 | 39.8 | 41.1 | 46.1 | 87.5 |
| Baichuan-13B | Dense | 0.00% | 61.5 | 36.4 | 91.5 | 65.8 | 49.0 | 64.2 | 69.1 | 78.2 | 48.1 | 46.0 | 54.8 | 55.3 | 60.0 | 100 |
| | Ours (None) | 24.7% | 48.8 | 34.8 | 50.2 | 62.2 | 40.4 | 46.4 | 56.7 | 68.2 | 30.6 | 27.7 | 52.9 | 55.1 | 47.8 | 79.7 |
| | Ours (FFN) | 25.5% | 58.3 | 35.1 | 77.5 | 64.1 | 36.5 | 57.7 | 58.2 | 69.4 | 26.5 | 28.8 | 53.1 | 54.3 | 51.6 | 86.0 |
| | Ours (Layer) | 24.7% | 59.1 | 36.1 | 83.7 | 62.0 | 36.5 | 58.2 | 59.4 | 71.8 | 27.8 | 25.0 | 52.3 | 56.1 | 52.3 | 87.2 |

Table 16: Stability of different pruning methods on classification benchmarks by pruning Baichuan-7B and Baichuan-13B.

| LLM | Method | Ratio | Benchmarks | | | | | | | | | | | | Average |
| --- | --- | --- | C3 | CMNLI | CHID | BoolQ | WSC | CoQA | HeSW | PIQA | Race-M | Race-H | MMLU | CMMLU | |
| Baichuan-7B | LLMPruner | 24.2% | 70.2 | 40.0 | 71.4 | 24.9 | 91.4 | 75.3 | 75.9 | 82.4 | 68.8 | 66.8 | 54.7 | 53.6 | 64.6 |
| | Ours (None) | 24.2% | 55.5 | 45.1 | 31.3 | 76.8 | 93.3 | 64.7 | 50.1 | 61.0 | 68.9 | 65.9 | 74.0 | 60.8 | 62.3 |
| | Ours (FFN) | 25.1% | 84.1 | 77.8 | 75.6 | 65.4 | 93.3 | 76.5 | 83.1 | 84.3 | 77.1 | 70.3 | 74.2 | 74.6 | 78.0 |
| | Ours (Layer) | 24.2% | 86.3 | 79.3 | 82.3 | 40.7 | 93.3 | 77.6 | 81.3 | 83.5 | 75.8 | 71.0 | 75.9 | 75.6 | 76.9 |
| Baichuan-13B | Ours (None) | 24.7% | 67.2 | 75.8 | 54.1 | 66.6 | 51.0 | 74.9 | 74.6 | 78.5 | 52.3 | 55.2 | 82.7 | 89.3 | 68.5 |
| | Ours (FFN) | 25.5% | 85.7 | 87.3 | 82.1 | 81.3 | 43.3 | 81.2 | 81.7 | 79.8 | 46.6 | 61.1 | 84.3 | 83.7 | 74.8 |
| | Ours (Layer) | 24.7% | 88.6 | 92.7 | 89.3 | 72.9 | 43.3 | 83.5 | 86.1 | 88.2 | 49.2 | 52.1 | 83.4 | 90.4 | 76.6 |

Table 17: Accuracy of different pruning methods on generation benchmarks by pruning Baichuan-7B and Baichuan-13B.

| LLM | Method | Ratio | Benchmarks | | | Average | RP |
| --- | --- | --- | Xsum | GSM8K | StrategyQA | | |
| Baichuan-7B | Dense | 0.00% | 19.1 | 9.84 | 55.5 | 28.1 | 100 |
| | LLMPruner | 24.2% | 12.6 | 1.74 | 40.4 | 18.2 | 64.8 |
| | Ours (None) | 24.2% | 0.3 | 0 | 0 | 0.1 | 0 |
| | Ours (FFN) | 25.1% | 19.3 | 2.11 | 41.1 | 20.8 | 74.0 |
| | Ours (Layer) | 24.2% | 18.2 | 1.36 | 38.7 | 19.4 | 69.0 |
| Baichuan-13B | Dense | 0.00% | 24.6 | 27.1 | 61.1 | 37.6 | 100 |
| | Ours (None) | 24.7% | 2.1 | 1.2 | 12.3 | 5.2 | 13.8 |
| | Ours (FFN) | 25.5% | 23.1 | 2.1 | 47.3 | 24.2 | 64.4 |
| | Ours (Layer) | 24.7% | 22.2 | 2.4 | 43.2 | 22.6 | 60.1 |

### E.3 EXPERIMENTAL RESULTS OF BAICHUAN2-7B AND BAICHUAN2-13B

We conduct experiments on Baichuan2-7B and Baichuan2-13B. The experimental results are shown in Table 18, Table 19 and Table 20. The results indicate that our proposed LLM-Streamline is superior to the concurrent SOTA method, LaCo.

Table 18: Accuracy of different pruning methods on classification benchmarks by pruning Baichuan2-7B and Baichuan2-13B. "*" indicates that we refer to the results in the original paper.

| LLM | Method | Ratio | Benchmarks | | | | | | | | | | | | Average | RP |
| --- | --- | --- | C3 | CMNLI | CHID | BoolQ | WSC | CoQA | HeSW | PIQA | Race-M | Race-H | MMLU | CMMLU | | |
| Baichuan2-7B | Dense | 0.00% | 64.4 | 33.4 | 85.5 | 63.1 | 42.3 | 63.1 | 67.6 | 76.1 | 51.1 | 52.5 | 54.7 | 57.1 | 59.2 | 100 |
| | LLMPruner | 24.2% | 39.9 | 33.9 | 70.6 | 50.0 | 42.3 | 38.7 | 52.7 | 70.4 | 22.3 | 22.8 | 24.9 | 24.9 | 41.1 | 69.4 |
| | LaCo* | 24.2% | 50.9 | 33.0 | 76.2 | 56.2 | 42.3 | 47.3 | 52.3 | 68.5 | 27.7 | 29.0 | 31.5 | 31.2 | 45.5 | 76.9 |
| | Ours (None) | 24.2% | 45.7 | 33.0 | 58.0 | 62.6 | 36.5 | 41.9 | 46.3 | 62.4 | 25.6 | 27.4 | 43.0 | 46.5 | 44.1 | 74.5 |
| | Ours (FFN) | 25.1% | 58.2 | 33.0 | 74.1 | 61.2 | 36.5 | 47.6 | 54.3 | 68.0 | 29.1 | 30.5 | 52.1 | 56.7 | 50.1 | 84.6 |
| | Ours (Layer) | 24.2% | 60.4 | 34.9 | 72.2 | 62.7 | 36.5 | 48.8 | 52.5 | 67.0 | 35.5 | 36.8 | 54.0 | 56.3 | 51.5 | 87.0 |
| Baichuan2-13B | Dense | 0.00% | 65.6 | 33.2 | 86.7 | 66.8 | 42.3 | 65.6 | 71.1 | 78.1 | 68.9 | 67.2 | 59.6 | 61.3 | 63.9 | 100 |
| | LaCo* | 24.7% | 61.1 | 33.0 | 76.7 | 62.4 | 44.2 | 55.5 | 60.7 | 68.9 | 57.8 | 56.9 | 51.4 | 53.7 | 56.9 | 89.0 |
| | Ours (None) | 24.7% | 59.1 | 34.4 | 81.9 | 61.8 | 36.5 | 53.9 | 61.9 | 71.0 | 63.0 | 60.4 | 50.3 | 57.9 | 57.7 | 90.3 |
| | Ours (FFN) | 25.5% | 63.0 | 33.0 | 81.7 | 60.1 | 36.5 | 54.7 | 62.1 | 70.5 | 71.1 | 68.2 | 57.1 | 58.2 | 59.7 | 93.4 |
| | Ours (Layer) | 24.7% | 63.5 | 33.0 | 84.1 | 62.0 | 38.5 | 56.9 | 63.0 | 72.0 | 70.2 | 66.3 | 59.1 | 60.2 | 60.7 | 95.0 |

Table 19: Stability of different pruning methods on classification benchmarks by pruning Baichuan2-7B and Baichuan2-13B.

| LLM | Method | Ratio | Benchmarks | | | | | | | | | | | | Average |
| | | | C3 | CMNLI | CHID | BoolQ | WSC | CoQA | HeSW | PIQA | Race-M | Race-H | MMLU | CMMLU | |
|---|---|---|---|---|---|---|---|---|---|---|---|---|---|---|---|
| Baichuan2-7B | LLMPruner | 24.2% | 62.2 | 50.6 | 75.1 | 55.5 | 63.5 | 66.0 | 80.3 | 87.0 | 54.2 | 51.7 | 50.0 | 46.9 | 61.9 |
| | Ours (None) | 24.2% | 68.5 | 98.1 | 63.8 | 69.1 | 84.6 | 61.3 | 67.3 | 72.0 | 50.6 | 48.0 | 69.3 | 74.5 | 68.9 |
| | Ours (FFN) | 25.1% | 81.1 | 98.1 | 77.1 | 68.2 | 84.6 | 66.5 | 78.1 | 77.3 | 61.2 | 57.7 | 87.3 | 89.2 | 77.2 |
| | Ours (Layer) | 24.2% | 83.3 | 92.8 | 74.9 | 67.6 | 84.6 | 68.0 | 77.2 | 80.9 | 64.1 | 62.9 | 89.2 | 88.6 | 77.8 |
| Baichuan2-13B | Ours (None) | 24.7% | 85.0 | 88.7 | 86.5 | 85.1 | 82.7 | 79.5 | 82.2 | 85.1 | 84.6 | 83.0 | 74.2 | 85.6 | 83.5 |
| | Ours (FFN) | 25.5% | 86.4 | 99.0 | 87.2 | 84.7 | 82.7 | 77.6 | 83.2 | 85.7 | 83.2 | 81.1 | 90.2 | 91.7 | 86.1 |
| | Ours (Layer) | 24.7% | 87.9 | 99.0 | 89.0 | 87.1 | 84.7 | 80.2 | 84.9 | 86.9 | 89.4 | 87.0 | 91.7 | 92.5 | 88.4 |

Table 20: Accuracy of different pruning methods on generation benchmarks by pruning Baichuan2-7B and Baichuan2-13B. "*" indicates that we refer to the results in the original paper.

| LLM | Method | Ratio | Benchmarks | | | Average | RP |
| | | | Xsum | GSM8K | StrategyQA | | |
|---|---|---|---|---|---|---|---|
| Baichuan2-7B | Dense | 0.00% | 21.0 | 24.8 | 60.0 | 35.3 | 100 |
| | LLMPruner | 24.2% | 14.5 | 1.4 | 10.8 | 8.9 | 25.2 |
| | LaCo* | 24.2% | 12.0 | - | - | - | - |
| | Ours (None) | 24.2% | 12.1 | 1.7 | 30.7 | 14.8 | 41.9 |
| | Ours (FFN) | 25.1% | 15.9 | 2.7 | 37.1 | 18.6 | 52.7 |
| | Ours (Layer) | 24.2% | 16.8 | 2.3 | 34.8 | 18.0 | 51.0 |
| Baichuan2-13B | Dense | 0.00% | 25.3 | 53.2 | 65.9 | 48.1 | 100 |
| | LaCo* | 24.7% | 12.3 | - | - | - | - |
| | Ours (None) | 24.7% | 17.2 | 3.3 | 37.2 | 19.2 | 39.9 |
| | Ours (FFN) | 25.5% | 21.3 | 3.1 | 48.8 | 24.4 | 50.7 |
| | Ours (Layer) | 24.7% | 20.9 | 5.5 | 51.3 | 25.9 | 53.8 |

## E.4 EXPERIMENTAL RESULTS OF LLAMA3.1-8B AND LLAMA3.1-70B

We conduct experiments on Llama3.1-8B and Llama3.1-70B. The experimental results are shown in Table 21 and Table 22. The results indicate that our proposed LLM-Streamline is superior to the previous SOTA method.

Table 21: Accuracy of different pruning methods on classification benchmarks by pruning Llama3.1-8B and Llama3.1-70B.

| LLM | Method | Ratio | Benchmarks | | | | | | | | | | | | Average | RP |
| | | | C3 | CMNLI | CHID | BoolQ | WSC | CoQA | HeSW | PIQA | Race-M | Race-H | MMLU | CMMLU | | |
|---|---|---|---|---|---|---|---|---|---|---|---|---|---|---|---|---|
| Llama3.1-8B | Dense | 0.00% | 65.3 | 33.0 | 73.8 | 68.2 | 36.5 | 69.8 | 74.7 | 81.1 | 71.6 | 64.5 | 66.8 | 52.5 | 63.2 | 100 |
| | SliceGPT | 23.9% | 38.4 | 32.1 | 21.3 | 37.8 | 38.5 | 38.0 | 39.9 | 58.7 | 21.9 | 23.3 | 25.8 | 25.2 | 33.4 | 52.8 |
| | Ours (None) | 24.4% | 42.3 | 33.7 | 19.3 | 52.3 | 36.5 | 30.7 | 28.4 | 58.9 | 36.6 | 33.3 | 39.1 | 34.4 | 36.9 | 58.4 |
| | Ours (Layer) | 24.4% | 55.9 | 34.5 | 54.5 | 67.6 | 36.5 | 62.5 | 62.6 | 74.5 | 64.8 | 55.9 | 64.9 | 51.5 | 57.1 | 90.6 |
| Llama3.1-70B | Dense | 0.00% | 74.8 | 33.0 | 81.6 | 76.5 | 37.5 | 73.0 | 79.9 | 83.9 | 86.8 | 80.5 | 79.3 | 68.8 | 71.3 | 100 |
| | SliceGPT | 29.1% | 40.4 | 31.9 | 18.9 | 37.8 | 37.5 | 41.0 | 45.3 | 61.0 | 24.1 | 24.8 | 37.5 | 30.5 | 35.9 | 50.4 |
| | Ours (None) | 30.3% | 66.1 | 37.5 | 58.1 | 69.0 | 46.2 | 61.8 | 68.4 | 75.7 | 81.7 | 73.2 | 70.4 | 62.0 | 64.2 | 90.0 |
| | Ours (Layer) | 30.3% | 68.9 | 34.7 | 70.0 | 72.5 | 42.3 | 68.9 | 74.4 | 79.3 | 86.8 | 81.5 | 78.6 | 68.2 | 68.8 | 96.5 |

Table 22: Stability of different pruning methods on classification benchmarks by pruning Llama3.1-8B and Llama3.1-70B.

| LLM | Method | Ratio | Benchmarks | | | | | | | | | | | | Average |
| | | | C3 | CMNLI | CHID | BoolQ | WSC | CoQA | HeSW | PIQA | Race-M | Race-H | MMLU | CMMLU | |
|---|---|---|---|---|---|---|---|---|---|---|---|---|---|---|---|
| Llama3.1-8B | SliceGPT | 23.9% | 63.2 | 37.0 | 38.5 | 35.6 | 98.1 | 60.7 | 59.3 | 71.2 | 39.0 | 46.1 | 42.1 | 47.4 | 53.2 |
| | Ours (None) | 24.4% | 59.9 | 47.0 | 39.0 | 57.0 | 100 | 49.8 | 43.4 | 64.7 | 53.4 | 54.5 | 57.8 | 61.4 | 53.7 |
| | Ours (Layer) | 24.4% | 78.5 | 49.7 | 59.9 | 75.0 | 100 | 80.3 | 84.8 | 86.5 | 87.3 | 86.1 | 90.8 | 89.1 | 80.7 |
| Llama3.1-70B | SliceGPT | 29.1% | 55.6 | 45.4 | 32.1 | 36.1 | 98.1 | 58.8 | 59.6 | 72.0 | 32.9 | 39.6 | 48.7 | 45.9 | 49.3 |
| | Ours (None) | 30.3% | 77.6 | 43.6 | 64.5 | 73.0 | 91.4 | 76.8 | 84.6 | 85.0 | 92.4 | 89.3 | 84.1 | 81.4 | 78.6 |
| | Ours (Layer) | 30.3% | 86.7 | 95.7 | 76.1 | 77.7 | 95.2 | 89.5 | 92.6 | 93.4 | 97.2 | 96.3 | 95.9 | 94.6 | 90.9 |

### E.5 EXPERIMENTAL RESULTS OF MIXTRAL-8X7B-V0.1

We conduct experiments on Mixture of Experts(MoE) model Mixtral-8x7B-v0.1. The experimental results are shown in Table 23 and Table 24.

Table 23: Accuracy of LLM-Streamline on classification benchmarks by pruning Mixtral-8x7B-v0.1.

| LLM | Method | Ratio | Benchmarks | | | | | | | | | | | | Average | RP |
| | | | C3 | CMNLI | CHID | BoolQ | WSC | CoQA | HeSW | PIQA | Race-M | Race-H | MMLU | CMMLU | | |
| --- | --- | --- | --- | --- | --- | --- | --- | --- | --- | --- | --- | --- | --- | --- | --- | --- |
| Mixtral-8x7B-v0.1 | Dense | 0.00% | 54.1 | 33.0 | 48.6 | 68.4 | 56.7 | 68.2 | 76.2 | 81.7 | 72.4 | 70.9 | 71.3 | 52.8 | 62.9 | 100.0 |
| | Ours (None) | 24.9% | 39.0 | 33.0 | 26.9 | 62.8 | 39.4 | 45.6 | 55.4 | 70.2 | 41.3 | 43.3 | 67.7 | 39.2 | 47.0 | 74.7 |
| | Ours (Layer) | 24.9% | **51.2** | **34.4** | **41.5** | **66.3** | **56.7** | **62.0** | **68.3** | **77.9** | **54.5** | **55.7** | **69.9** | **50.2** | **57.4** | **91.3** |

Table 24: Stability of LLM-Streamline on classification benchmarks by pruning Mixtral-8x7B-v0.1.

| LLM | Method | Ratio | Benchmarks | | | | | | | | | | | | Average |
| | | | C3 | CMNLI | CHID | BoolQ | WSC | CoQA | HeSW | PIQA | Race-M | Race-H | MMLU | CMMLU | |
| --- | --- | --- | --- | --- | --- | --- | --- | --- | --- | --- | --- | --- | --- | --- | --- |
| Mixtral-8x7B-v0.1 | Ours (None) | 24.9% | 66.9 | **98.9** | 62.4 | 76.4 | 38.5 | 69.3 | 72.4 | 80.6 | 57.0 | 60.3 | 79.1 | 64.4 | 68.9 |
| | Ours (Layer) | 24.9% | **83.6** | 87.6 | **76.2** | **81.3** | **100** | **86.8** | **90.4** | **90.8** | **67.8** | **69.0** | **85.3** | **80.4** | **83.3** |

### E.6 EXPERIMENTAL RESULTS OF OPT-1.3B AND OPT-2.7B

We also conduct experiments on small models (OPT-1.3B and OPT-2.7B). The experimental results are shown in Table 25. The results indicate that our proposed LLM-Streamline is superior to the previous SOTA method, across different pruning rates.

Table 25: Accuracy of different pruning methods by pruning OPT-1.3B and OPT-2.7B.

| LLM | Method | Ratio | Benchmarks | | | | | | Average | RP |
| | | | PIQA | WinoGrande | HellaSwag | ARC-easy | ARC-challenge | OpenBookQA | | |
| --- | --- | --- | --- | --- | --- | --- | --- | --- | --- | --- |
| OPT-1.3B | Dense | 0.00% | 72.4 | 59.3 | 53.7 | 51.0 | 29.5 | 23.4 | 48.2 | 100.0 |
| | SliceGPT | 18.1% | 67.6 | 53.6 | 35.7 | 51.1 | 23.1 | 20.2 | 41.9 | 86.9 |
| | Ours(None) | 19.4% | 57.2 | 51.7 | 29.1 | 32.5 | 22.7 | 13.2 | 34.4 | 71.4 |
| | Ours(FFN) | 18.1% | **68.8** | **58.4** | **39.1** | **54.3** | **23.3** | **23.3** | **44.5** | **92.3** |
| | Dense | 0.00% | 72.4 | 59.3 | 53.7 | 51.0 | 29.5 | 23.4 | 48.2 | 100.0 |
| | SliceGPT | 25.8% | 65.5 | 52.8 | 34.2 | 48.8 | 24.4 | 17.0 | 40.5 | 84.0 |
| | Ours(None) | 27.1% | 52.2 | 51.1 | 25.7 | 26.6 | 20.5 | 14.0 | 31.7 | 65.8 |
| | Ours(FFN) | 25.8% | **66.4** | **56.0** | **36.8** | **51.6** | 22.2 | **21.0** | **42.3** | **87.8** |
| | Dense | 0.00% | 72.4 | 59.3 | 53.7 | 51.0 | 29.5 | 23.4 | 48.2 | 100.0 |
| | SliceGPT | 33.6% | 62.4 | 52.6 | 32.2 | 45.4 | **23.1** | 16.6 | 38.7 | 80.3 |
| | Ours(None) | 34.8% | 50.5 | 51.5 | 25.8 | 26.2 | 20.3 | 14.6 | 31.5 | 65.4 |
| | Ours(FFN) | 33.6% | **62.9** | 52.1 | **33.9** | **48.3** | 20.8 | **20.6** | **39.8** | **82.6** |
| OPT-2.7B | Dense | 0.00% | 73.8 | 61.0 | 45.9 | 60.9 | 26.8 | 25.0 | 48.9 | 100.0 |
| | SliceGPT | 16.8% | 69.6 | 56.3 | 40.4 | 56.2 | **27.5** | 20.2 | 45.0 | 92.0 |
| | Ours(None) | 17.8% | 61.2 | 54.1 | 33.8 | 41.2 | 24.1 | 15.8 | 38.4 | 78.5 |
| | Ours(FFN) | 16.8% | **70.7** | **60.4** | **42.9** | **57.8** | 25.3 | **24.4** | **46.9** | **95.9** |
| | Dense | 0.00% | 73.8 | 61.0 | 45.9 | 60.9 | 26.8 | 25.0 | 48.9 | 100.0 |
| | SliceGPT | 25.7% | **69.1** | 55.0 | 37.9 | 53.9 | **26.7** | 18.2 | 43.5 | 89.0 |
| | Ours(None) | 26.7% | 59.7 | 53.4 | 33.5 | 38.1 | 24.3 | 15.4 | 37.4 | 76.5 |
| | Ours(FFN) | 25.7% | 67.0 | **59.5** | **40.3** | **54.6** | 24.7 | **22.2** | **44.7** | **91.4** |
| | Dense | 0.00% | 73.8 | 61.0 | 45.9 | 60.9 | 26.8 | 25.0 | 48.9 | 100.0 |
| | SliceGPT | 34.6% | 64.8 | 54.1 | 35.6 | 50.0 | **26.5** | 18.0 | 41.5 | 84.9 |
| | Ours(None) | 35.6% | 56.6 | 52.9 | 31.5 | 37.6 | 24.1 | 14.9 | 36.3 | 74.2 |
| | Ours(FFN) | 34.6% | **65.3** | **55.3** | **36.3** | **51.4** | 24.5 | **21.0** | **42.3** | **86.5** |

### E.7 EXPERIMENTAL RESULTS OF LLAMA2-7B AT AROUND 50% PRUNING RATIO

We conduct experiments on Llama2-7B at a higher pruning ratio of approximately 50%. The experimental results are shown in Table 26 and Table 27. The results indicate that our proposed LLM-Streamline is superior to the previous SOTA method.

Table 26: Accuracy of different pruning methods on classification benchmarks by pruning Llama2-7B at a higher pruning ratio of approximately 50%.

| LLM | Method | Ratio | C3 | CMNLI | CHID | BoolQ | WSC | CoQA | HeSW | PIQA | Race-M | Race-H | MMLU | CMMLU | Average | RP |
|---|---|---|---|---|---|---|---|---|---|---|---|---|---|---|---|---|
| | Dense | 0.00% | 43.8 | 33.0 | 41.6 | 70.8 | 37.5 | 66.7 | 71.3 | 78.1 | 33.1 | 35.5 | 46.8 | 31.8 | 49.2 | 100.0 |
| | LLMPruner | 49.2% | 26.4 | 33.2 | 17.3 | 43.5 | 38.5 | 25.4 | 32.7 | 59.2 | 22.3 | 21.7 | 23.4 | 24.8 | 30.7 | 63.4 |
| Llama2-7B | SliceGPT | 48.3% | 26.5 | 32.1 | 15.4 | 38.1 | 42.3 | 28.1 | 30.9 | 53.6 | 23.6 | 23.1 | 25.2 | 25.3 | 30.4 | 61.8 |
| | Ours (None) | 48.0% | 33.1 | 34.0 | 17.3 | 55.4 | 36.5 | 31.0 | 34.3 | 56.3 | 26.8 | 27.2 | 34.9 | 27.9 | 34.6 | 70.3 |
| | Ours (Layer) | 48.0% | 39.7 | 33.0 | 27.7 | 62.1 | 36.5 | 44.3 | 45.2 | 63.6 | 24.6 | 24.3 | 33.2 | 28.1 | 38.5 | 78.3 |

Table 27: Stability of different pruning methods on classification benchmarks by pruning Llama2-7B at a higher pruning ratio of approximately 50%.

| LLM | Method | Ratio | C3 | CMNLI | CHID | BoolQ | WSC | CoQA | HeSW | PIQA | Race-M | Race-H | MMLU | CMMLU | Average |
|---|---|---|---|---|---|---|---|---|---|---|---|---|---|---|---|
| | LLMPruner | 49.2% | 63.6 | 96.2 | 60.8 | 33.8 | 87.5 | 51.5 | 55.5 | 70.9 | 47.8 | 48.1 | 56.9 | 58.1 | 60.9 |
| Llama2-7B | SliceGPT | 48.3% | 48.0 | 34.9 | 54.7 | 19.0 | 83.7 | 57.7 | 52.7 | 63.4 | 71.2 | 61.5 | 52.2 | 58.1 | 54.8 |
| | Ours (None) | 48.0% | 73.2 | 63.0 | 62.2 | 65.2 | 95.2 | 52.2 | 52.5 | 62.6 | 48.2 | 52.5 | 61.4 | 58.8 | 62.3 |
| | Ours (Layer) | 48.0% | 80.8 | 100 | 67.7 | 79.5 | 95.2 | 65.3 | 66.1 | 76.0 | 58.7 | 57.3 | 63.8 | 59.4 | 72.5 |

## E.8 RESULTS OF SUFFICIENT POST-TRAINING

Following the method outlined in Section 4.7, We conduct experiments using the entire SlimPajama-6B for post-training, and the results are presented in Table 28. As shown, using the entire dataset resulted in a slight improvement, but at a significant computational cost, requiring 100 times the computational time.

Table 28: Detailed accuracy results with different training data volumes.

| LLM | Method | Training data size | C3 | CMNLI | CHID | BoolQ | WSC | CoQA | HeSW | PIQA | Race-M | Race-H | MMLU | CMMLU | Xsum | GSM8k | StrategyQA | Average | RP |
|---|---|---|---|---|---|---|---|---|---|---|---|---|---|---|---|---|---|---|---|
| | Dense | - | 43.8 | 33.0 | 41.6 | 70.8 | 37.5 | 66.7 | 71.3 | 78.1 | 33.1 | 35.5 | 46.8 | 31.8 | 19.4 | 16.5 | 60.2 | 45.7 | 100.0 |
| Llama2-7B | Layer-First | 30k | 43.9 | 33.0 | 29.8 | 70.8 | 36.5 | 59.6 | 64.3 | 73.4 | 36.6 | 37.4 | 44.9 | 30.0 | 19.7 | 2.05 | 54.8 | 42.5 | 93.0 |
| | Layer-First | 5.49M | 43.5 | 33.0 | 33.2 | 68.8 | 46.2 | 61.1 | 66.5 | 76.0 | 31.8 | 29.9 | 47.3 | 31.8 | 18.2 | 10.6 | 58.6 | 43.8 | 95.8 |

## E.9 DETAILED RESULTS OF DIFFERENT LIGHTWEIGHT NETWORKS

The detailed results of accuracy and stability of different lightweight networks on different classification benchmarks are shown in Table 29 and Table 30. We can observe that FFN achieves the best results. Meanwhile, for the Transformer Layer, inheritance of the pruned first layer yields the best results.

Table 29: Detailed accuracy results of different lightweight networks on different classification benchmarks, where "†" indicates that the intermediate size of the added lightweight network is half that of the default LLM's intermediate size.

| LLM | Method | Ratio | C3 | CMNLI | CHID | BoolQ | WSC | CoQA | HeSW | PIQA | Race-M | Race-H | MMLU | CMMLU | Average | RP |
|---|---|---|---|---|---|---|---|---|---|---|---|---|---|---|---|---|
| | Dense | 0.00% | 43.8 | 33.0 | 41.6 | 70.8 | 37.5 | 66.7 | 71.3 | 78.1 | 33.1 | 35.5 | 46.8 | 31.8 | 49.2 | 100.0 |
| | Layer-Random | 24.0% | 42.5 | 33.0 | 27.0 | 65.9 | 36.5 | 58.0 | 58.8 | 70.2 | 35.0 | 36.0 | 46.3 | 31.4 | 45.1 | 91.7 |
| | Layer-First | 24.0% | 43.3 | 33.0 | 24.1 | 67.5 | 36.5 | 59.2 | 61.1 | 71.5 | 34.8 | 37.0 | 45.5 | 29.4 | 45.2 | 91.9 |
| | Layer-Last | 24.0% | 43.5 | 33.0 | 29.0 | 64.5 | 41.4 | 56.8 | 61.5 | 71.6 | 34.5 | 35.0 | 46.0 | 30.8 | 45.6 | 92.7 |
| Llama2-7B | Layer-Avg | 24.0% | 42.1 | 33.0 | 26.7 | 66.3 | 36.5 | 57.7 | 59.4 | 69.9 | 34.3 | 34.5 | 43.3 | 28.5 | 44.4 | 90.2 |
| | FFN† | 26.0% | 40.6 | 33.0 | 24.2 | 67.5 | 36.5 | 58.4 | 59.5 | 71.4 | 41.9 | 41.4 | 46.3 | 30.8 | 46.0 | 93.5 |
| | FFN | 25.0% | 40.7 | 33.0 | 22.8 | 65.9 | 38.5 | 60.6 | 61.2 | 71.2 | 38.0 | 38.7 | 47.0 | 31.7 | 45.8 | 93.1 |
| | SwiGLU† | 26.0% | 41.9 | 33.0 | 24.3 | 68.5 | 36.5 | 55.8 | 57.9 | 69.6 | 29.9 | 33.3 | 43.4 | 31.6 | 43.8 | 89.0 |
| | SwiGLU | 25.0% | 40.9 | 33.0 | 22.1 | 67.0 | 36.5 | 56.9 | 59.1 | 70.0 | 33.8 | 35.0 | 45.6 | 30.8 | 44.2 | 89.8 |

Table 30: Detailed stability results of different lightweight networks on different classification benchmarks, where "†" indicates that the intermediate size of the added lightweight network is half that of the default LLM's intermediate size.

| LLM | Method | Ratio | Benchmarks | | | | | | | | | | | | Average |
| | | | C3 | CMNLI | CHID | BoolQ | WSC | CoQA | HeSW | PIQA | Race-M | Race-H | MMLU | CMMLU | |
| --- | --- | --- | --- | --- | --- | --- | --- | --- | --- | --- | --- | --- | --- | --- | --- |
| Llama2-7B | Layer-Random | 24.0% | 79.9 | 100 | 68.2 | 83.8 | **95.2** | 77.4 | 82.0 | 83.2 | 69.9 | 71.9 | 85.1 | 77.4 | 81.2 |
| | Layer-First | 24.0% | 79.8 | 100 | 64.4 | **86.3** | **95.2** | **81.7** | 85.3 | 85.6 | 81.8 | 79.0 | 82.4 | 71.0 | 82.7 |
| | Layer-Last | 24.0% | **81.0** | 100 | **73.5** | 84.9 | 82.7 | 81.3 | **85.8** | 85.4 | 82.3 | 76.3 | 83.0 | 66.9 | 81.9 |
| | Layer-Avg | 24.0% | 80.7 | 100 | 68.5 | 84.4 | **95.2** | 78.9 | 82.8 | 82.5 | 73.0 | 70.7 | 74.6 | 58.8 | 79.2 |
| | FFN† | 26.0% | 79.9 | 100 | 65.4 | 82.1 | **95.2** | 78.7 | 80.7 | 81.7 | 74.7 | 70.3 | 84.9 | 74.6 | 80.7 |
| | FFN | 25.0% | 79.8 | 100 | 64.1 | 83.1 | 93.3 | 80.7 | 84.7 | 84.6 | 85.1 | 79.0 | 87.5 | 82.5 | **83.7** |
| | SwiGLU† | 26.0% | 78.5 | 100 | 64.5 | 78.9 | **95.2** | 75.4 | 80.9 | 82.1 | **89.3** | **87.1** | 80.7 | 78.4 | 82.6 |
| | SwiGLU | 25.0% | 78.9 | 100 | 63.5 | 84.4 | **95.2** | 77.0 | 82.2 | 82.3 | 85.7 | 82.4 | 87.9 | 79.7 | 83.3 |

## E.10 DETAILED RESULTS OF DIFFERENT PRUNING RATIO

The detailed results of accuracy and stability on LLMs under different pruning ratios on different classification benchmarks are shown in Table 31 and Table 32. The experiment results show that the performance of the pruned model is linearly correlated with the number of parameters, demonstrating the effectiveness of our method.

Table 31: Detailed accuracy results of different pruning ratios on different classification benchmarks, where "†" indicates that the intermediate size of the added lightweight network is half that of the default LLM's intermediate size.

| LLM | Method | Ratio | Benchmarks | | | | | | | | | | | | Average | RP |
| | | | C3 | CMNLI | CHID | BoolQ | WSC | CoQA | HeSW | PIQA | Race-M | Race-H | MMLU | CMMLU | | |
| --- | --- | --- | --- | --- | --- | --- | --- | --- | --- | --- | --- | --- | --- | --- | --- | --- |
| Llama2-7B | Dense | 0.00% | 43.8 | 33.0 | 41.6 | 70.8 | 37.5 | 66.7 | 71.3 | 78.1 | 33.1 | 35.5 | 46.8 | 31.8 | 49.2 | 100.0 |
| | FFN† | 2.0% | **43.6** | **33.0** | **41.0** | **71.5** | **36.5** | **65.4** | **70.7** | **77.5** | 36.1 | 38.1 | 46.3 | **31.6** | **49.3** | **100.0** |
| | FFN† | 11.0% | 42.1 | **33.0** | 35.1 | 69.0 | **37.5** | 63.4 | 67.8 | 75.6 | 35.2 | 37.0 | **46.5** | 29.9 | 47.7 | 97.0 |
| | FFN† | 20.0% | 42.3 | **33.0** | 29.0 | 70.2 | **36.5** | 62.7 | 63.8 | 72.3 | 37.8 | 37.7 | 45.7 | 28.8 | 46.7 | 94.9 |
| | FFN† | 26.0% | 40.6 | **33.0** | 24.2 | 67.5 | **36.5** | 58.4 | 59.5 | 71.4 | **41.9** | **41.4** | 46.3 | 30.8 | 46.0 | 93.5 |
| TinyLlama-1.1B | Dense | 0.00% | 38.3 | 34.6 | 30.4 | 56.4 | 47.1 | 48.8 | 54.5 | 71.3 | 24.1 | 25.8 | 25.8 | 25.0 | 40.2 | 100.0 |
| OpenLlama-3B-v2 | Dense | 0.00% | 43.0 | 33.0 | 31.1 | 60.6 | 37.5 | 58.7 | 65.3 | 77.0 | 25.1 | 26.9 | 27.0 | 25.3 | 42.5 | 100.0 |

Table 32: Detailed stability results of different pruning ratios on different classification benchmarks, where "†" indicates that the intermediate size of the added lightweight network is half that of the default LLM's intermediate size.

| LLM | Method | Ratio | Benchmarks | | | | | | | | | | | | Average |
| | | | C3 | CMNLI | CHID | BoolQ | WSC | CoQA | HeSW | PIQA | Race-M | Race-H | MMLU | CMMLU | |
| --- | --- | --- | --- | --- | --- | --- | --- | --- | --- | --- | --- | --- | --- | --- | --- |
| Llama2-7B | FFN† | 2.0% | **95.8** | 100 | **93.2** | 86.5 | 95.2 | 96.2 | 97.8 | 97.1 | 93.0 | 89.5 | **95.5** | 89.7 | **94.1** |
| | FFN† | 11.0% | 90.4 | 100 | 82.5 | 91.4 | 94.2 | 90.2 | 93.4 | 92.7 | 89.6 | 82.7 | 84.6 | 72.0 | 88.6 |
| | FFN† | 20.0% | 81.6 | 100 | 71.9 | 86.0 | 95.2 | 84.4 | 87.3 | 87.9 | 85.2 | 78.6 | 78.1 | 67.7 | 83.7 |
| | FFN† | 26.0% | 79.9 | 100 | 65.4 | 82.1 | 95.2 | 78.7 | 80.7 | 81.7 | 74.7 | 70.3 | 84.9 | 74.6 | 80.7 |

## E.11 DETAILED RESULTS OF LAYER REPLACEMENT AND LORA

The detailed results of accuracy and stability of different layer replacement strategies and LoRA on different benchmarks are shown in Table 33 and Table 34. The results show that layer replacement outperforms LoRA in both accuracy and stability.

Table 33: Detailed accuracy results of layer replacement and LoRA on different classification benchmarks, where "†" indicates that the intermediate size of the added lightweight network is half that of the default LLM's intermediate size.

| LLM | Method | Ratio | Benchmarks | | | | | | | | | | | | Average | RP |
|---|---|---|---|---|---|---|---|---|---|---|---|---|---|---|---|---|
| | | | C3 | CMNLI | CHID | BoolQ | WSC | CoQA | HeSW | PIQA | Race-M | Race-H | MMLU | CMMLU | | |
| Llama2-7B | Dense | 0.00% | 43.8 | 33.0 | 41.6 | 70.8 | 37.5 | 66.7 | 71.3 | 78.1 | 33.1 | 35.5 | 46.8 | 31.8 | 49.2 | 100.0 |
| | Layer-First | 24.0% | 43.9 | **33.0** | 29.8 | **70.8** | 36.5 | 59.6 | 64.3 | 73.4 | 36.6 | 37.4 | 44.9 | 30.0 | 46.7 | 94.9 |
| | Layer-Last | 24.0% | **44.9** | **33.0** | 29.4 | 69.2 | 36.5 | 58.9 | 63.5 | 74.1 | 37.8 | 37.8 | **46.5** | 30.4 | **46.8** | **95.1** |
| | Layer-Avg | 24.0% | 43.9 | **33.0** | 30.1 | 67.5 | 36.5 | 58.3 | 62.5 | 72.3 | 36.6 | 36.1 | 46.2 | **31.9** | 46.2 | 93.9 |
| | FFN† | 26.0% | 41.6 | **33.0** | 25.8 | 62.6 | 36.5 | 58.9 | 62.1 | 72.3 | **41.9** | 40.2 | 44.4 | 30.5 | 45.8 | 93.1 |
| | FFN | 25.0% | 43.8 | **33.0** | 27.0 | 68.7 | 36.5 | 60.7 | 63.5 | 72.4 | 37.4 | 35.4 | 45.3 | 31.5 | 46.3 | 94.1 |
| | SwiGLU† | 26.0% | 44.0 | **33.0** | 27.9 | 61.2 | 36.5 | 57.2 | 61.7 | 71.2 | 30.3 | 32.9 | 45.0 | **31.9** | 44.4 | 90.2 |
| | SwiGLU | 25.0% | 43.2 | **33.0** | 27.1 | 67.0 | 36.5 | 58.2 | 62.1 | 71.2 | 35.1 | 35.7 | 45.8 | 30.8 | 45.5 | 92.5 |
| | LoRA | 24.0% | 43.2 | **33.0** | 27.6 | 63.5 | 36.5 | 57.7 | 62.4 | 71.7 | 30.7 | 32.9 | 43.5 | 30.8 | 44.5 | 90.4 |

Table 34: Detailed stability results of layer replacement and LoRA on different classification benchmarks, where "†" indicates that the intermediate size of the added lightweight network is half that of the default LLM's intermediate size.

| LLM | Method | Ratio | Benchmarks | | | | | | | | | | | | Average |
|---|---|---|---|---|---|---|---|---|---|---|---|---|---|---|---|
| | | | C3 | CMNLI | CHID | BoolQ | WSC | CoQA | HeSW | PIQA | Race-M | Race-H | MMLU | CMMLU | |
| Llama2-7B | Layer-First | 24.0% | 82.7 | **100** | 74.3 | 84.7 | 95.2 | 85.7 | 89.1 | 88.6 | 84.1 | 82.2 | 84.5 | 77.2 | **85.7** |
| | Layer-Last | 24.0% | 83.2 | **100** | 73.8 | 87.9 | 95.2 | 89.5 | 85.8 | **88.7** | 84.7 | 82.1 | 83.1 | 72.9 | 85.6 |
| | Layer-Avg | 24.0% | 81.0 | **100** | 72.3 | 67.0 | 95.2 | 84.1 | 87.4 | 86.6 | 87.3 | 82.2 | **90.5** | 73.7 | 83.9 |
| | FFN† | 26.0% | 82.0 | **100** | 71.4 | 75.4 | 95.2 | 82.9 | 86.6 | 87.2 | 79.0 | 75.3 | 85.1 | 81.0 | 83.4 |
| | FFN | 25.0% | 80.6 | **100** | 72.0 | 83.5 | 95.2 | 85.4 | 87.7 | 87.2 | 84.5 | 81.0 | 85.4 | 79.3 | 85.2 |
| | SwiGLU† | 26.0% | 80.0 | **100** | 71.4 | 63.1 | 95.2 | 80.5 | 86.1 | 85.1 | **90.4** | **87.7** | 89.2 | **87.7** | 84.7 |
| | SwiGLU | 25.0% | 81.6 | **100** | 72.9 | 67.3 | 95.2 | 81.8 | 86.5 | 85.3 | 87.7 | 84.8 | 90.0 | 83.3 | 84.7 |
| | LoRA | 24.0% | 81.9 | **100** | 73.4 | 59.1 | 95.2 | 81.6 | 84.8 | 85.3 | 85.2 | 81.3 | 82.1 | 75.3 | 82.1 |

# F  LIMITATION

Our method achieves SOTA results compared to existing pruning methods, but its performance still falls short of other commonly used model compression methods, e.g., quantization. Therefore, we plan to enhance the performance of our pruning method and explore combining it with other compression and inference acceleration techniques to make it more practical.

