# OpenReview forum: "Streamlining Redundant Layers to Compress Large Language Models"
_ICLR.cc/2025/Conference — ICLR 2025 Spotlight_

### Official Review · Reviewer_MF5H · 2024-10-29

**Soundness:** 2
**Presentation:** 2
**Contribution:** 2
**Rating:** 6
**Confidence:** 4

**Summary:**

This work addresses the compression of LLMs through layer pruning and layer replacement. A lightweight module is trained to substitute for the pruned layers. Additionally, a new evaluation metric is proposed to mitigate the limitations of accuracy-based metrics.

**Strengths:**

* The concept of layer replacement for performance recovery is intriguing.
* I appreciate the detailed comparison between perplexity and cosine similarity as pruning metrics, effectively revealing the data-sensitive drawbacks of perplexity.
* The paper comprehensively covers various types of LLMs and evaluation benchmarks.
* I like the exploration of multiple designs for lightweight networks.
* Overall, the paper is well-written and easy to follow.

**Weaknesses:**

* I think that replacing layers with FFN or Transformer layers may not provide the full inference speed benefits typically gained through layer pruning, even if parameter counts remain similar. Could you analyze the speedup of layer replacement and compare it with pure layer pruning?
* It’s unclear how to set the number of contiguous layers (n) in Equation (3). How many layers are typically removed at once during pruning? Could you elaborate on this aspect?
* Is this method intended as one-shot pruning (pruning-replacement once) or iterative pruning (pruning-replacement, then pruning-replacement again)? Would it be possible to compare the two approaches?
* Why is MSE chosen as the loss function for the replacement module in Equation (3)? What would happen if it were trained using the original language model loss?
* A 25% compression rate seems relatively modest. Could you provide a comparison at higher pruning ratios (e.g., 50% layer removal) against baseline methods?

**Questions:**

Please see the above weakness section.

---

> ### Author Response · Authors · 2024-11-13
>
> Thank you for your thorough review and the constructive feedback on our manuscript. We have carefully reviewed and considered each of the questions and suggestions you have raised. Below, we provide detailed responses to your comments. In the content below, we use **W** to represent Weakness and **Q** to represent Question.
>
>
> **W1: Comparison of the acceleration effect of layer replacement with pure layer pruning.**
>
> We are sorry for the misunderstanding. In our experiments, under the same sparsity constraint, when using a Transformer layer as the lightweight network, the pruned model's structure remains identical to that of pure layer pruning. As a result, the acceleration effect of layer replacement is equivalent to that of pure layer pruning. While when using an MLP as the lightweight network, our pruned model includes a layer that omits the attention architecture, resulting in a slightly better acceleration effect compared to pure layer pruning.
>
>
> We also report the inference speed (in output tokens per second) in **Table 11** of the appendix in the revised paper (We have also put this table at the bottom of comments.) . The results indicate that our approach (layer) — replacing with Transformer layer— achieves the same speed (25.68 tokens/s) as our approach (None) — pure layer pruning. Additionally, our approach (FFN) — replacing with MLP — is slightly faster, at 25.88 tokens/s, compared to pure layer pruning.
>
> **W2: Regarding how many layers are removed at once in Equation 3.**
>
> Thank you for your question. The number of layers removed is determined based on the specified sparsity. For instance, in our experiments, we set the sparsity to about 25%, meaning we prune layers equivalent to 25% of the total parameters at once. For example, Llama2-7B has 32 layers, so pruning 25% of the model's parameters is roughly equivalent to removing 8 layers.
>
> **W3: one-shot pruning or iterative pruning**
>
> We are sorry for the misunderstanding. All of our experiments use the one-shot pruning approach that you mentioned. Iterative pruning—removing one layer at a time and replacing it with a lightweight network—would not significantly reduce the overall number of parameters. This is because a lightweight network, such as an MLP, contains roughly two-thirds the parameters of a single Transformer layer. Therefore, replacing each layer individually with a separate lightweight network, rather than using one lightweight network to replace multiple layers, would not lead to a significant reduction in parameters. On Llama2-7B, to achieve approximately 25% sparsity by using different lightweight networks to replace each layer, we would need to train about 24 lightweight networks to replace 24 Transformer layers. This results in a significantly high training cost.
>
> **W4: Comparison between MSE loss and the original language model loss**
>
> Thank you for raising the comment! In terms of computational resource usage, training with MSE loss only requires storing the lightweight network in memory, without needing to load the entire LLM. In contrast, training with the original language model loss requires loading the full LLM into memory. By using MSE loss, we minimize the memory requirements during training, making LLM-Streamline feasible in memory-constrained environments.
>
> **W5: Experiments on higher sparsity level**
>
> Thank you for your suggestion. We have conducted experiments at higher sparsity levels and included the results in **Appendix E.7** of the revised paper. These results demonstrate that LLM-Streamline remains significantly stronger than the baseline method, even at a high pruning ratio of approximately 50%.
>
>
>
> |          Llama2-7B         | Dense | LLM-Pruner | SliceGPT | Ours(None) | Ours(FFN) | Ours(Layer) |
> |:--------------------------:|:-----:|:----------:|:--------:|:----------:|:---------:|:-----------:|
> |      Pruning Ratio (%)     |  0.00 |    24.8    |   25.4   |    24.0    |    25.0   |     24.0    |
> | Inference Speed (tokens/s) | 19.87 |    25.91   |   27.20     |    25.68   |   25.88   |    25.68    |
>
> Table 1: The inference speed of models pruned using different methods.

---

> ### Author Response · Authors · 2024-11-24
>
> Dear Reviewer MF5H,
>
> Thank you for taking the time to review our paper. We sincerely appreciate your valuable feedback and have provided detailed responses and additional experiments to address your concerns. Kindly let us know if you have any further questions. Your insights are invaluable to us and will undoubtedly help us further improve the quality of our work.
>
> Once again, thank you for your time and thoughtful consideration.
>
> Best
>
> The Authors

---

> > ### Comment · Reviewer_MF5H · 2024-11-25
> >
> > Apologies for my delayed response, and thank you for the comprehensive rebuttal. I appreciate the report on inference time reduction and the clear explanation of the pruning scheme.
> >
> > I have also reviewed the rationale for using MSE loss, which is primarily for practical reasons, and I find it understandable. I am quite surprised by the robust results achieved with a high pruning ratio and am thankful for the additional experiments.
> >
> > Initially, my review score was somewhat harsh, due to uncertainties about the method's effectiveness and efficiency gains. However, the authors have addressed my concerns effectively, and the paper presents many interesting empirical findings. Therefore, I would like to increase my score from 3 to 6.
> >
> > One final question: Do the authors plan to release the code? While this will not affect my review score, releasing the codebase would help verify reproducibility and encourage further research.

---

> > > ### Author Response · Authors · 2024-11-25
> > >
> > > Dear Reviewer MF5H,
> > >
> > > Thank you for your recognition of our work and response. We will release the code as soon as the anonymity period concludes.
> > >
> > > Best
> > >
> > > The Authors

---

### Official Review · Reviewer_M4rG · 2024-10-30

**Soundness:** 3
**Presentation:** 4
**Contribution:** 3
**Rating:** 8
**Confidence:** 4

**Summary:**

The paper presents LLM-Streamline, a novel layer-wise pruning and replacement framework aimed at compressing large language models (LLMs) by identifying and removing less significant layers. The process is divided into two stages: (1) Layer Pruning to identify and prune layers with minimal impact based on cosine similarity, and (2) Layer Replacement, which trains a lightweight network to compensate for the removed layers, mitigating performance loss. Additionally, a new stability metric is introduced to more accurately reflect performance post-pruning, which is compared against existing methods across multiple benchmarks.

**Strengths:**

* This paper uses a lightweight network and training based on hidden states before and after compression to compensate for the loss caused by pruning, reducing the need for computing resources and leading to better precision recovery.

* The continuous layer pruning used in this paper reduces the complexity of the compressed model and is easier to accelerate on hardware than unstructured pruning and other methods.

* The paper provides an in-depth analysis of the limitations within traditional accuracy metrics, offering a well-thought-out solution to address it. Additionally, the ablation studies are exceptionally comprehensive, with thorough and insightful analysis throughout.

**Weaknesses:**

* The sparsity levels explored in the paper do not exceed 25%, leaving higher sparsity scenarios untested. In contrast, methods like LLMPruner evaluate and compare performance at a 50% sparsity level. This omission raises concerns about whether the proposed contiguous layer pruning approach would remain effective at higher sparsity levels.

* The models selected for evaluation are all based on the LLaMA architecture, which limits the assessment of the proposed method’s generalizability. Testing on more diverse architectures (e.g., MoE, GQA) and larger models with greater sparsity requirements (such as LLaMA-30B, LLaMA-70B) would provide a more comprehensive validation of LLM-Streamline’s effectiveness across various model types and scales.

* The study can add comparisons of real hardware performance (e.g., inference speed, FLOPs) with other structured pruning methods, where actual hardware constraints play a critical role in model selection and pruning effectiveness.

**Questions:**

Please refer to weaknesses section for questions.

---

> ### Author Response · Authors · 2024-11-13
>
> Thank you very much for your thorough review and valuable feedback on our manuscript. We have carefully reviewed and considered each of the questions and suggestions you have raised. Below, we provide detailed responses to your comments. In the content below, we use **W** to represent Weakness and **Q** to represent Question.
>
> **W1 and W2: Experiments on higher sparsity level, more diverse architectures, and larger models**
>
> Thank you for your suggestion. We will conduct experiments at higher sparsity levels, with more diverse architectures, and on larger models, and we will strive to update the paper with these results before the end of the rebuttal period.
>
> **W3: Result of real hardware performance**
>
> Thank you for your feedback. We have included the results for inference speed in the Appendix of the revised paper(See **Table 11**). We have also put this table at the bottom of comments.
>
>
>
> |          Llama2-7B         | Dense | LLM-Pruner | SliceGPT | Ours(None) | Ours(FFN) | Ours(Layer) |
> |:--------------------------:|:-----:|:----------:|:--------:|:----------:|:---------:|:-----------:|
> |      Pruning Ratio (%)     |  0.00 |    24.8    |   25.4   |    24.0    |    25.0   |     24.0    |
> | Inference Speed (tokens/s) | 19.87 |    25.91   |   27.20     |    25.68   |   25.88   |    25.68    |
>
> Table 1: The inference speed of models pruned using different methods.

---

> ### Author Response · Authors · 2024-11-24
>
> Dear Reviewer M4rG,
>
> We have conducted experiments with Llama2-7B at a 50% pruning rate, as well as additional experiments on Llama3.1-8B, Llama3.1-70B, and Mixtral-8x7B. The results are detailed in **Appendices E.4, E.5, and E.7** of revised paper. As demonstrated, our proposed method, LLM-Streamline, consistently outperforms the baseline approaches.
>
> Best
>
> The Authors

---

> > ### Comment · Reviewer_M4rG · 2024-11-24
> >
> > Thanks for your careful rebuttal. My concerns are well addressed in this rebuttal. Thus, I keep my original score to accept this paper.

---

> > > ### Author Response · Authors · 2024-11-25
> > >
> > > Dear Reviewer M4rG,
> > >
> > > Thank you for your recognition of our work and response.
> > >
> > > Best
> > >
> > > The Authors

---

### Official Review · Reviewer_wDBW · 2024-10-31

**Soundness:** 3
**Presentation:** 3
**Contribution:** 4
**Rating:** 8
**Confidence:** 5

**Summary:**

In this work, the authors propose LLM-Streamline, a new approach for compressing large language models (LLMs) by pruning layers based on cosine similarity and replacing the pruned layers with a lightweight network to preserve or even enhance model performance. This two-step method is shown to effectively reduce model size with minimal loss in accuracy, as demonstrated by experiments across various classification and generation benchmarks. Additionally, the work introduces a stability metric to address limitations of traditional accuracy metrics in evaluating pruned models. Experimental results highlight the superiority of LLM-Streamline over other state-of-the-art pruning methods in terms of accuracy retention and computational efficiency, especially under hardware constraints. Overall, this work offers a novel technique with practical implications for large-scale LLM deployments.

**Strengths:**

1. Originality: This paper combines layer pruning with lightweight network replacement in a novel approach for compressing LLMs. This method effectively maintains model performance even after significant pruning.
2. Significance: The proposed stability metric enhances LLM compression evaluation by addressing limitations in standard accuracy metrics, providing a potentially more reliable measure of retained model performance.
3. Technical Quality: The experiments are comprehensive, covering various benchmarks that showcase the model’s effectiveness in classification and content generation. Each step of the pruning and replacement process is clearly explained, highlighting the technical soundness of the approach.
4. Practical Implications: LLM-Streamline improves model efficiency while retaining performance, making it valuable for real-world applications where computational resources are limited.

**Weaknesses:**

1. Metric Justification: While cosine similarity is chosen as the primary metric for layer redundancy, additional justification for this choice over other metrics (e.g., perplexity, Euclidean distance) would be beneficial.
2. Comparison with Other Methods: While the authors mention alternative approaches, such as LoRA, they do not provide a detailed comparison. A more thorough discussion of how LLM-Streamline performs relative to other popular fine-tuning and compression techniques would give readers a clearer perspective.
3. Explanation of Stability Metric: The paper could simplify its explanation of the stability metric, making it more accessible to readers new to model compression. Clarifying why stability offers practical advantages over accuracy would also be beneficial.

**Questions:**

1. Can you provide more detail on why cosine similarity was chosen as the primary measure for layer redundancy? How does it compare to using perplexity or other metrics?
2. For the stability metric, is there a specific scenario or task where it offers clear advantages over traditional accuracy measures? An illustrative example would help clarify this.
3. Have you considered a detailed comparison with other model compression methods, like LoRA? This would help highlight what sets LLM-Streamline apart.

---

> ### Author Response · Authors · 2024-11-13
>
> Thank you very much for your thorough review and valuable feedback on our manuscript. We have carefully reviewed and considered each of the questions and suggestions you have raised. Below, we provide detailed responses to your comments. In the content below, we use **W** to represent Weakness and **Q** to represent Question.
>
>
> **W1 and Q1: Comparison of Cosine Similarity with Other Metrics**
>
> Thanks for your question. In **lines 159-176** of our paper, we detail our rationale for selecting cosine similarity over other metrics. Since most modern LLMs utilize a post-norm architecture, this results in the norm of the hidden states increasing with the depth of the layers. This leads to a bias where deeper layers tend to exhibit higher dot product similarity, while earlier layers show smaller Euclidean distances.
>
>
> Regarding perplexity, we find it to be a highly sensitive metric. As shown in our experiments (see **lines 756-783** in the appendix), using perplexity as a pruning metric leads to the selection of different pruning layers depending on the pre-training dataset. Moreover, we compare the performance of models pruned using perplexity with those pruned using cosine similarity. The results suggest that models pruned using cosine similarity outperform those pruned with perplexity.
>
>
>
> **W2 and Q3: Comparison of LLM-Streamline and LoRA**
>
> Thanks for your suggestion. In **lines 199-215** of our paper, we highlight the advantages of LLM-Streamline over LoRA. Specifically, LLM-Streamline offers lower memory consumption and faster training speeds. Moreover, the lightweight network used in LLM-Streamline is trained through distillation, resulting in enhanced performance compared to LoRA. Furthermore, in **lines 473-485**, we present a detailed comparison between LLM-Streamline and LoRA, with experimental results showing that LLM-Streamline consistently outperforms LoRA in both performance and resource efficiency.
>
>
> **W3: A simplified explanation of stability**
>
> Thank you for your suggestion. We can easily explain our motivation for introducing the concept of stability. After pruning, a model may correctly answer questions that it previously answered incorrectly in classification tasks based on perplexity (PPL), which can even lead to an increase in accuracy for certain tasks. However, this does not necessarily indicate that the model’s overall capability has improved after pruning. Therefore, we propose stability as a metric, which focuses on the consistency of the model’s responses before and after pruning. We think it can provides a more accurate reflection of the pruned model’s performance than accuracy alone.
>
>
> **Q2: Examples that illustrate the advantage of stability over accuracy**
>
> Thank you for your valuable suggestion. For example, in PIQA, there is a question like this:
>
> The following makes sense:
>
> A.
> Question: How to make tissue paper window decorations? Answer: Find tissue paper and fold it in half. Take scissors and cut out pieces of the paper in the middle. When you are done, tape it to your window.
>
> B.
> Question: How to make tissue paper window decorations? Answer: Find tissue paper and fold it in half. Take scissors and tear out pieces of the paper in the middle. When you are done, tape it to your window.
>
>
> Llama2-7B has a very similar PPL for both options, indicating great uncertainty in this question and choosing the wrong answer B. However, the pruned Llama2-7B still has a very similar PPL for both options but manages to guess the correct answer A. This situation reflects the limitations of accuracy, whereas stability can well identify such cases, thus more accurately reflecting the performance of the pruned model.

---

> > ### Comment · Reviewer_wDBW · 2024-11-29
> >
> > Thank you for your thoughtful and thorough revisions. Your responses have addressed my concerns raised in the initial review. The added explanations, comparative analyses, and practical examples have significantly improved the clarity and rigor of the manuscript. I therefore maintain my initial score to accept this work.

---

> > > ### Author Response · Authors · 2024-11-29
> > >
> > > Dear Reviewer wDBW,
> > >
> > > Thank you for your recognition of our work and response.
> > >
> > > Best regards,
> > >
> > > The Authors

---

### Official Review · Reviewer_YVX4 · 2024-11-03

**Soundness:** 3
**Presentation:** 3
**Contribution:** 3
**Rating:** 8
**Confidence:** 1

**Summary:**

This submission presented LLM-Streamline, a new model pruning method for LLMs. Besides traditional pruning, this method also proposed layer replacement, a novel module that trains a lightweight network to replace the pruned layers to mitigate performance loss. In addition, the authors also proposed stability as a new metric. Experimental results show the superiority of this method.

**Strengths:**

Basically, this submission has two major innovations:
1. layer replacement
2. new metric named stability.

The first one is a very good contribution that mitigates the loss of pruning only.

**Weaknesses:**

I didn't find any weakness of this submission.

**Questions:**

I'm not an expert of LLM pruning/compression field. I'm curious why you didn't compare with some knowledge distillation method for LLM, for example
https://proceedings.mlr.press/v235/ko24c.html

---

> ### Author Response · Authors · 2024-11-13
>
> Thank you very much for your thorough review and valuable feedback on our manuscript.
>
>
> In response to your question, “Why didn’t you compare with some knowledge distillation methods for LLM?”, we appreciate your insight. While both distillation and pruning are model compression techniques, they serve different purposes. Distillation is primarily used to improve the training of smaller models, whereas pruning focuses on directly reducing the size of a large model. Although some pruning methods, such as the LLM-Streamline approach we proposed, involve the training process and can incorporate distillation loss during this phase, the two methods can complement each other. Therefore, papers introducing distillation methods, like DistiLLM, typically do not compare with pruning techniques, and vice versa.

---

### Author Response · Authors · 2024-11-27

We sincerely appreciate the reviewers' valuable and constructive feedback on our manuscript. We are especially pleased to receive positive comments, including:
- **All reviewers recognized the novelty and effectiveness of our layer replacement method, as well as the proposed stability metrics.**
- **Reviewers wDBW, M4rG, and MF5H commended the comprehensiveness of our experiments.**
- **Reviewer wDBW highlighted the practical significance of our method for deploying large-scale LLMs under hardware constraints.**

The reviewers' insights were instrumental in improving our paper. We have carefully addressed all concerns and conducted additional experiments based on their suggestions. In response to the feedback, we have made the following revisions:
﻿
- **Additional Experiments**: We added results for high sparsity levels (**Appendix E.7**), evaluations on Llama3.1-8B and Llama3.1-70B (**Appendix E.4**), Mixtral-8x7B (**Appendix E.5**), and comparisons of inference speed (**Appendix D**).
- **References in Main Text**: The additional experiment results are now referenced in **Section 4.4** of the main text for better alignment.
- **Minor Adjustments**: To comply with the page limit, we made minor textual edits that do not affect the readability or content of the paper.

Thank you again for your time and consideration.

Best regards,

The Authors

---

### Meta-Review · Area_Chair_mKAV · 2024-12-20

**Metareview:**

The authors propose a method for layer pruning and so-called replacement in LLMs to minimise the impact of pruning. They also consider a dedicated metric to asses the resulting model compression. The claims are substantiated experimentally and were considered convincing by all reviewers. The authors also produced additional experimental results as requested by the reviewers during the rebuttal. The proposed method is of practical importance and of interest to the community.

**Additional Comments On Reviewer Discussion:**

Reviewers acknowledged the response provided by the authors. The clarifications provided by the authors were appreciated and let to the most critical reviewer to increase their scores. All reviewers agreed that this is a solid and novel piece of work.

---

### Decision · Program_Chairs · 2025-01-22

Accept (Spotlight)